# pH- and concentration-dependent supramolecular assembly of a fungal defensin plectasin variant into helical non-amyloid fibrils

Christin Pohl [1,2,6] ✉, Gregory Effantin [3], Eaazhisai Kandiah [4], Sebastian Meier [2], Guanghong Zeng[5,7], Werner Streicher[1,8], Dorotea Raventos Segura[1], Per H. Mygind[1,9], Dorthe Sandvang[1,10], Line Anker Nielsen[1], Günther H. J. Peters [2], Guy Schoehn [3], Christoph Mueller-Dieckmann[4], Allan Noergaard [1] & Pernille Harris [2,11] ✉

Self-assembly and fibril formation play important roles in protein behaviour. Amyloid fibril formation is well-studied due to its role in neurodegenerative diseases and characterized by refolding of the protein into predominantly β-sheet form. However, much less is known about the assembly of proteins into other types of supramolecular structures. Using cryo-electron microscopy at a resolution of 1.97 Å, we show that a triple-mutant of the anti-microbial peptide plectasin, PPI42, assembles into helical non-amyloid fibrils. The in vitro anti-microbial activity was determined and shown to be enhanced compared to the wildtype. Plectasin contains a cysteine-stabilised α-helix-β-sheet structure, which remains intact upon fibril formation. Two protofilaments form a right-handed protein fibril. The fibril formation is reversible and follows sigmoidal kinetics with a pH- and concentration dependent equilibrium between soluble monomer and protein fibril. This high-resolution structure reveals that α/β proteins can natively assemble into fibrils.

[1] Novozymes A/S, Bagsvaerd, Denmark. [2] Technical University of Denmark, Department of Chemistry, Kgs. Lyngby, Denmark. [3] Univ. Grenoble Alpes, CNRS, CEA, Institute for Structural Biology, F-38000 Grenoble, France. [4] European Synchrotron Radiation Facility, Grenoble, France. [5] DFM A/S (Danish National Metrology Institute), Hoersholm, Denmark. [6] Present address: Department of Biochemistry and Structural Biology, Lund University, Lund, Sweden. [7] Present address: Novo Nordisk, Bagsvaerd, Denmark. [8] Present address: NanoTemper Technologies GmbH, Muenchen, Germany. [9] Present address: Ascendis Pharma A/S, Hellerup, Denmark. [10] Present address: Chr. Hansen A/S, Hoersholm, Denmark. [11] Present address: Department of Chemistry, University of Copenhagen, Copenhagen, Denmark. ✉email: christin.pohl@biochemistry.lu.se; phharris@chem.ku.dk

Protein self-assembly is a hallmark of biomacromolecules. For protein polymers and fibrils, an assembly can be highly ordered and can serve different purposes[1,2]. Protein fibrils have been found to template melanin polymerization as protection against cytotoxic damage[3], serve as protection against environmental stress[4,5], stabilize biofilms[6], and be involved in signal transduction[7]. Many polymeric protein assemblies found in nature are amyloid fibrils that are characterized by a distinct cross-β sheet motif[8]. Amyloids are best known for their role in neurodegenerative diseases such as Alzheimer's disease[9] or prion-related diseases like Creutzfeldt-Jakob disease and Bovine Spongiform Encephalopathy[10]. Recent studies have shown a strong link between anti-microbial peptides (AMPs) and amyloid proteins. Amyloid-β, which is known to form amyloid fibrils associated with Alzheimer's disease, has been found to have anti-microbial and antifungal activity linked to the innate immune system[11–13]. Several other amyloids or their fragments, including α-synuclein[14], islet amyloid polypeptide (IAPP)[15–17], tau protein[18], human prion protein[19], and endostatin[20] have been reported to show anti-microbial activity. Furthermore, many AMPs, including, lysozymes[21], protegrin-1[22], HAL-2[23], uperin 3.5[24,25], dermaseptin S9[26], Cn-AMP2[27], and longipin[28] have been found to form amyloid or amyloid-like structures. In a direct comparison, similar mechanisms of membrane disruption have been found for amyloids and AMPs[16,29]. Dysregulation of amyloid proteins[30,31], as well as AMPs[32–34], has been associated with diseases or other negative effects. Fibril formation has, however, also been suggested as a storage method in the design of long-acting drugs[25,35].

Self-assembling proteins show great potential for functional materials[36]. Supramolecular assembling peptides have served as an inspiration for the design of new peptides with the desired functionality[37–42]. The study of anti-microbial polymers has mainly focused on their synthetic design to overcome challenges in immunogenicity, off-target effects and serum instability and ensure their controlled release[43–45]. The most prominent self-assembling AMPs are the MAX peptides, a group of designed peptides, that form an injectable hydrogel by undergoing a transition from an unstructured soluble form to a self-assembling β-hairpin[46–53]. Through the introduction of different point mutations, these AMPs show an array of tuneable features in their self-assembly with pH being the most prominent stimulus. pH as an external stimulus to control self-assembly and release has been used also in other designed or natural peptides and in nanomaterials[54–62]. The polymers of natural AMPs have been mainly claimed to be of amyloid nature[23,24,63]. However, in recent years, several high-resolution structures revealed assemblies that differed from the typical cross-β assembly in amyloid fibrils[25,64–66]. A variety of different protein fibril structures have been discovered, including cross α-helical amyloid-like fibrils[25,64,65] and functional α-helical assemblies of the anti-microbial LL-37[66].

Plectasin is an anti-microbial peptide with a molecular weight of 4.4 kDa that shows antibiotic behaviour against Gram-positive bacteria[67] by binding lipid II and preventing its incorporation into the bacterial cell wall as part of the peptidoglycan[68]. Plectasin consists of 40 amino acids and shows well-defined secondary structural elements in form of an α-helix (M13-S21) and an antiparallel β-sheet (G28-A31; V36-C39)[67,69]. The structure of plectasin is stabilized by three disulfide bridges (C4-C30; C15-C37; C19-C39), which is typical for this type of defensins and leads to high conformational stability[70,71]. Plectasin proved to be especially active against Streptococci[67], whereas variants of plectasin such as NZ2114 also showed improved activity against Staphylococci[72] and prevent post-therapy relapse compared to conventional antibiotics[73]. Additionally, plectasin showed favourable characteristics such as low toxicity, high serum stability and long in vivo half-life[67,68] and appears to be a promising protein drug candidate in the efforts against increasing bacterial resistance. Plectasin variant NZ2114 has already been used in the development of a novel type of anti-microbial catheter, that had plectasin adsorbed into the material and subsequently released upon exposure to aqueous solutions[74]. In our previous study, we characterized the solution behaviour of the plectasin wildtype and three variants at acidic pH (pH 3.5–5.5)[70]. We found a pH-dependent loss in nuclear magnetic resonance (NMR) signal intensity of plectasin variant D9S Q14K V36L, referred to as PPI42, at constant protein concentration and an exponential increase of apparent molecular mass and radius of gyration $R_g$ with increasing protein concentration in small-angle X-ray scattering (SAXS) measurements, indicating that large protein clusters formed. Energy minimization after introducing mutations, however, predicted only minor effects on the overall structure of PPI42 compared to the plectasin wildtype (Supplementary Fig. 1)[70].

Here, we describe the reversible sequestration of PPI42 into helical fibrils using cryo-EM and biophysical techniques. We show that the introduced mutations improved anti-microbial potency against Staphylococci compared to wildtype and retained activity against Streptococci clinical isolates. PPI42 fibril structure is comprised of the native-like protein forming single protofilaments or a right-handed mature fibril consisting of two protofilaments. We characterize the pH and protein concentration-dependent monomer-fibril equilibrium. Our study displays an atomic-level characterization of fibrils in equilibrium with a sparsely soluble monomer formed by an α/β defensin. This study enhances our understanding of protein self-assembly and might inspire the design of new AMPs with desirable properties.

## Results

**Plectasin variant PPI42 forms a gel with a regular fibril superstructure.** Plectasin variant PPI42 formed a stiff hydrogel when diluted or dialyzed into neutral pH (here pH 7), whereas the plectasin wildtype remained in solution under the same conditions (Fig. 1a). We evaluated the elasticity of the hydrogel based on the ability to stay in the upper part of the test tube after turning and whether an introduced bubble will float to the top of the gel, similar to the assay performed by ref. [75]. We found the gel elasticity strongly depends on protein concentration. We tested several different buffers within this study (histidine, tris, acetate, citrate, or no buffer), but gel formation occurred irrespective of the chemical nature of these buffers. When the formed hydrogel was dialyzed against pH below 5, the gel formation proved reversible, and the gel dissolved into a liquid.

Using atomic force microscopy (AFM), the nanoscale structure of the gel formed at pH 5 was investigated at protein concentrations of ~20 mg/mL (4.5 mM), which resembles the protein concentration of the protein stock after dialysis. The measurements were performed using PeakForce Quantitative Nanomechanical Mapping (QNM^TM). This mode made simultaneous mapping of topography and elasticity of the sample possible. Long, pearl necklace-like fibrils could be observed (Fig. 1b). Different buffer systems were tested, but no significant differences in the topography were observed (Supplementary Fig. 2), suggesting that the buffer composition does not influence the fibril formation and that pH is the main promoting factor. The height of the fibrils in different buffers was measured at 3.8 ± 0.3 nm with a periodicity of 29 nm. All samples showed similar morphology and comparable elasticity of 0.7–1.0 GPa, indicating that the fibrils formed by PPI42 were possibly suitable for high-resolution structural studies by electron microscopy (EM).

Negative stain EM was used to gain more insights into the structure of the fibrils formed by PPI42. The fibrils were formed

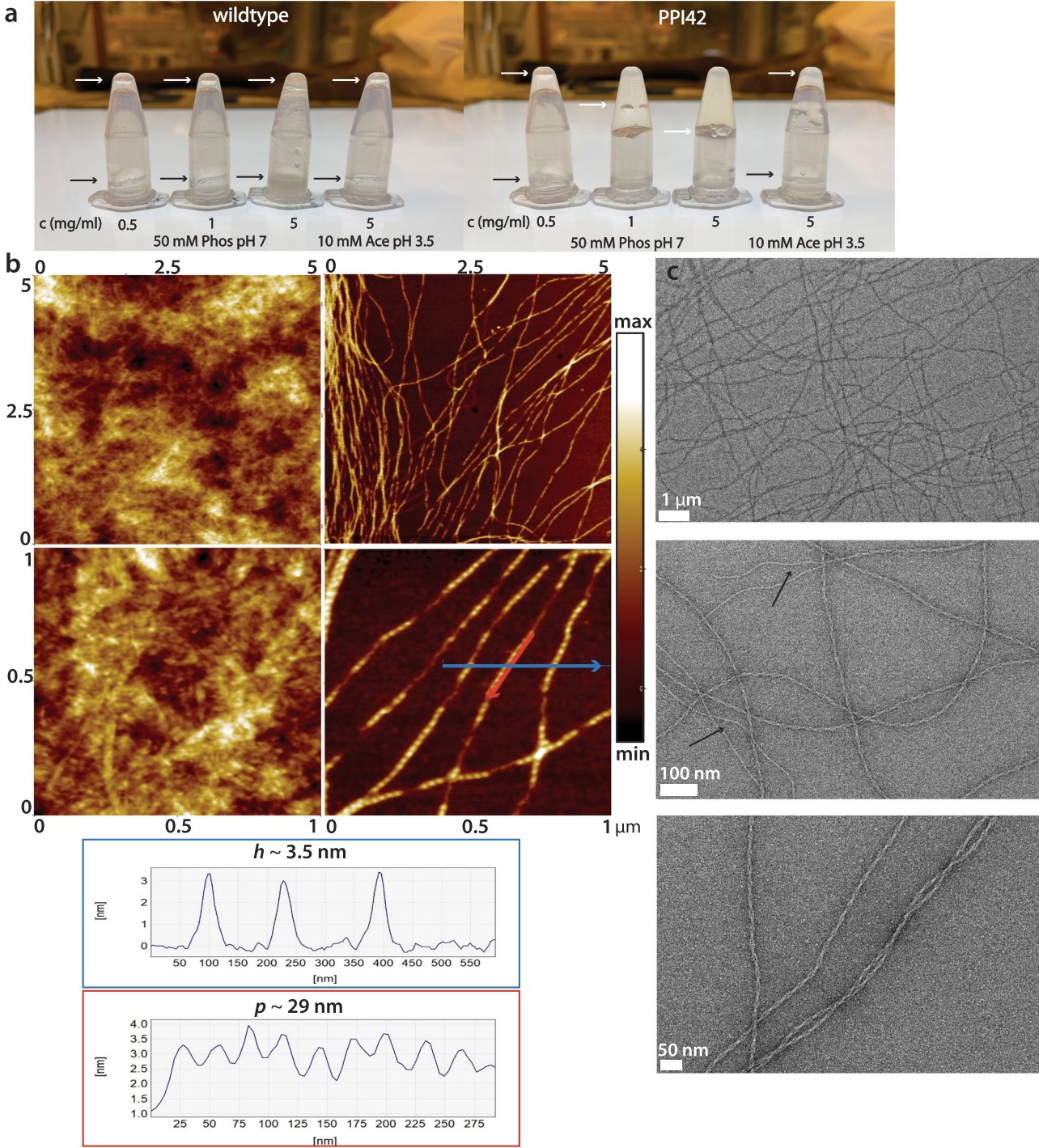

**Fig. 1 Elasticity and topography of the hydrogel formed by PPI42. a** Gel formation assay to assess the elasticity of the hydrogel formed by PPI42 (right) compared to plectasin wildtype (left). Samples were diluted to a concentration series of 0.5, 1 and 5 mg/mL (phosphate buffer pH 7). A control sample with c = 5 mg/mL (1.25 mM) was prepared by dilution into the dialysis buffer (acetate pH 3.5). Samples were assessed based upon, whether an introduced bubble was rising (white arrow) and whether samples solidified or dropped to the bottom (black arrow). **b** AFM height images of various ranges and line profiles measured along the shown arrows (acetate buffer, pH 5, $n = 2$, representative images shown). The line profile in the blue box corresponds to measurement of the fibril height $h$ along the blue arrow, the line profile in the red box corresponds to measurement of the periodicity $p$ along the red arrow. Figure axis indicate measurement dimensions (in μm). **c** Images of the fibrils using negative stain EM (acetate buffer, pH 5.5, $n = 2$, representative images shown). PPI42 fibrils formed protein fibrils consisting of two coiling protofilaments. Frayed fibril termini and single fibrils are indicated by arrows.

at pH 5.5, which we found more reproducible for the formation of the hydrogel, and the total protein concentration after dialysis was ~20 mg/mL (4.5 mM). EM images revealed that the pearl necklace-like structure derived from AFM resulted from two protofilaments winding around each other forming the mature

fibril (Fig. 1c). We observed frayed fibril termini and a small number of single protofilaments (Fig. 1c, black arrows), suggesting that the protofilaments formed first and then assembled into the mature fibrils. The network built by PPI42 fibrils showed similar features in AFM and negative stain EM

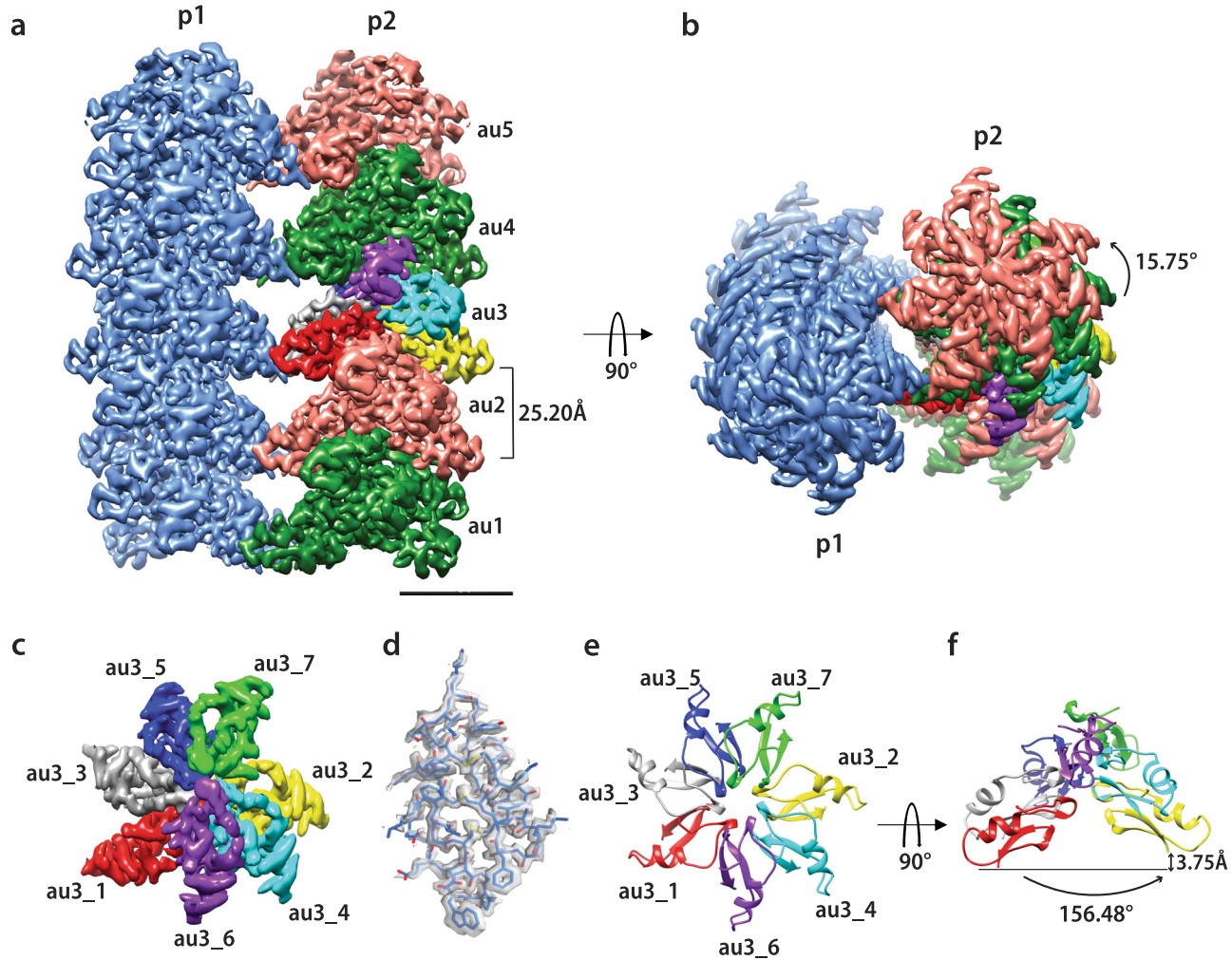

**Fig. 2 High-resolution cryo-EM structure of the protein fibril formed by PPI42. a, b** Isosurface representation of the cryo-EM map along the transverse axis (**a**) and along the longitudinal axis (**b**) of the fibril (90° rotation). The structure is composed of two coiling protofilaments (p1 and p2). The asymmetric units (au) are related by helical symmetry (axial rise of 25.20 Å and azimuthal angle of 15.75° shown for the p2 protofilament) and coloured differently (green and salmon). The axial rise (25.20 Å) (**a**) and the azimuthal angle (15.75°) (**b**) between two repeating units are indicated. Each asymmetric unit is composed of seven monomers. The scale bar is equivalent to 30 Å. **c** Isosurface of asymmetric unit au3 with each monomer coloured differently. View shown from the top (same orientation as in panel b). Each of the seven monomers composing the asymmetric unit are labelled (au3_1 to au3_7). **d** Atomic model of a PPI42 monomer (au3_3) refined into the cryo-EM coulomb potential map. **e, f** Atomic model of au3 viewed from the top (**e**) and the side (**f**). Each pair of monomers (such as au3_1 and au3_2) within an asymmetric unit are related to each other by a pseudo helical symmetry (average axial rise of 3.75 ± 0.15 Å and an average twist of 156.48 ± 2.25°).

(Fig. 1). The periodicity of 29 nm along the fibril obtained by negative stain EM (Supplementary Fig. 3) was in agreement with the periodicity measured with AFM.

**Cryo-EM structure of the supramolecular assembly**. We were able to determine the structure of the plectasin fibril at an overall resolution of 1.97 Å (local resolution varies between 1.9 and 3.0 Å (Supplementary Fig. 4)) using cryo-EM, which is one of the highest resolutions hitherto achieved for a protein fibril in cryo-EM and allowed an unambiguous structure determination (Fig. 2, Supplementary Fig. 4). The structure consists of two coiling protofilaments, which form a right-handed superstructure (mature fibril). Each protofilament has an axial (C2) and a helical symmetry (axial rise $z = 25.20$ Å and a helical twist phi = 15.75°) (Fig. 2a, b). The repeating or asymmetric unit (au) for each protofilament is composed of seven monomers. They are arranged in a near-helical way by forming a pseudo-right-handed helix with an average axial rise of 3.75 ± 0.15 Å and an average twist of 156.48 ± 2.25° between two consecutive monomers (Fig. 2f). The

axial rise also appeared, when measured by X-ray fibre diffraction, as a sharp diffraction ring at ~4 Å (Supplementary Fig. 5). Weaker diffraction rings were observed at 6.2, 8.0, and 9.5 Å, which correspond most likely to repeating distances between the monomers. The overall structure of PPI42 within the fibril proved to be similar to the plectasin wildtype and showed only minor differences to the simulated structure of PPI42 in solution (all-atom RMSD: 1.059 Å)[70]. The N-terminal loop and the loop connecting the two β-sheets were oriented toward the protofilament centre while the C-terminus and the loop between α-helix and β-sheet were facing outwards. The PPI42 protofilament is predominantly stabilized both by polar and hydrophobic interactions (Fig. 3). In total, each PPI42 monomer interacts with five other monomers (Fig. 2). The PPI42 protofilament is stabilized by a hydrophobic core in the centre (Fig. 3a, red spheres) and hydrophobic interfaces between the monomers (Fig. 3a, black spheres). The hydrophobic centre consists of residues (P7), (W8), and (F35) (Fig. 3b, Supplementary Fig. 4). CH/π-interactions between (P7) and (W8) additionally stabilizes the protofilament

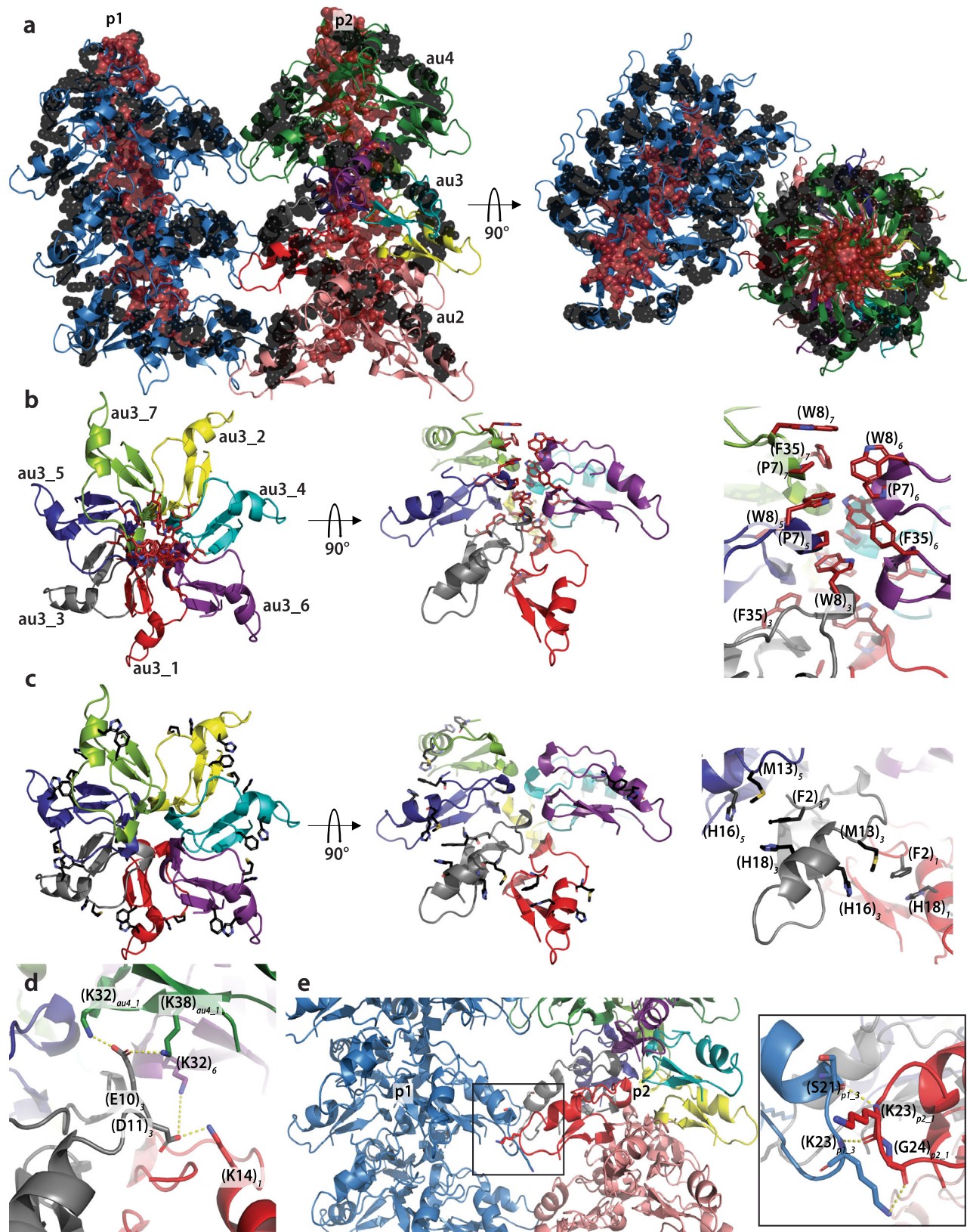

centre (Fig. 3b). Hydrophobic interactions on the fibril surface consist of (F2) and (H18), which stabilize each other by aromatic π-stacking, clustering with (M13) and (H16) on the adjacent molecule (Fig. 3c). This arrangement results in a series of interfaces that are located in a ring-like manner (Fig. 3a). Even though the local resolution is lower in that region (~2.5 Å), we could

conclude that the negatively charged sidechains of (E10) and (D11) appear to play an essential role in the arrangement of PPI42 in the protofilament by forming a network of salt bridges to different lysines (Fig. 3d). In one asymmetric unit composed of seven monomers each residue $(D11)_n$ forms a salt bridge to the $(K14)_{n-2}$ sidechain (numbering continues to adjacent subunits in

**Fig. 3 Stabilizing protein–protein interactions within mature fibrils of PPI42. a** Stabilizing, mostly hydrophobic clusters that form an outer ring, consisting of (F2), (M13), (H16) and (H18) (black spheres) and a hydrophobic core consisting of (P7), (W8) and (F35) (red spheres). View along the transverse axis (left) of the fibril and view along the longitudinal axis (right) of the fibril is shown (90° rotation). **b** Hydrophobic (red amino acids shown as sticks) interactions in the fibril centre shown in one asymmetric unit (au3, monomers coloured individually) with view from the top (left) and from the side (middle). Detailed arrangement of hydrophobic amino acids in the protofilament center is shown on the right. Numbering corresponds to the relative monomer position in the asymmetric unit. **c** Hydrophobic interactions (black amino acids shown as sticks) forming the outer hydrophobic ring of one protofilament shown in one asymmetric unit (au3) with view from the top (left) and from the side (middle). Detailed arrangement of hydrophobic amino acids on the protofilament surface id shown on the right. **d** Representative polar interactions of au3_3 (yellow lines) of acidic amino acids (E10) and (D11) with (K14) of the adjacent monomer au3_1 and with (K32) of au3_6 and (K32) and (K38) of monomers au4_1 above. **e** Polar interactions (yellow lines) of monomer 1 and 3 of each asymmetric unit (here au3) between the two protofilaments p1 and p2 forming the right-handed mature fibril with view along the longitudinal axis. (K23) and (G24) backbone of monomer 1 (here au3_1$_{p2}$) forms three polar interaction with (S21) backbone and (K23) backbone and sidechain of monomer 3 (here au3_3$_{p1}$).

the same manner). Additionally, $(D11)_n$ forms a salt bridge to $(K32)_{n+3}$. Residue $(E10)_n$ is also connected by forming a salt bridge to $(K32)_{n+6}$ and $(K38)_{n+6}$. On the opposite site $(N5)_n$ forms a hydrogen bond to the $(G6)_{n+2}$ backbone. At the centre of the protofilament $(W8)_n$ forms a hydrogen bond to the $(G33)_{n+3}$ backbone. The assembly of the two protofilaments into the superstructure is favoured by the interaction of the loop between α-helix and β-sheet of monomer 1 and 3 in each asymmetric unit with the two symmetric monomers in the other protofilament. The $(G24)_1$ and $(K23)_1$ backbone form three hydrogen bonds with the $(K23)_3$ sidechain and backbone and $(S21)_3$ backbone which stabilize the mature fibril (Fig. 3e). Additionally, we calculated the buried area of PPI42 upon fibril formation, which has been shown to correlate directly with the binding affinity between monomers[76]. We found that upon formation of the protofilament, the buried area increases from ~29% to ~57% with slight variation between the different monomers within the asymmetric unit (Supplementary Table 1). The interfaces within the protofilament correspond to ~2500 Å$^2$ per monomer. The interface between the two protofilaments was calculated to be 722 Å$^2$ per asymmetric unit, which corresponds to a small interface area for protein complexes[76]. However, repetitive interfaces in a protein fibril amplify the strength of a complex formation.

**Cryo-EM structure of the isolated protofilament.** In our negative stain EM measurement, we observed a small portion of fibrils existing as single protofilaments. In order to investigate whether there are structural rearrangements upon formation of the mature fibril, we analyzed the 3.4 Å resolution structure of PPI42 forming the single protofilament obtained by cryo-EM (Supplementary Figs. 4 and 6). The structure of the protofilament was determined using the structure of the mature fibril. PPI42 monomers show a very similar arrangement in the protofilament compared to the mature fibril (all-atom RMSD: 0.727 Å) (Supplementary Fig. 6). The PPI42 protofilament forms a straight right-handed helical structure having an axial rise of 3.76 Å and a helical twist of 156.5° between two monomers (Supplementary Fig. 6). These helical parameters are very close to the seven monomers of the asymmetric unit in the mature fibril (average axial rise of 3.75 ± 0.15 Å and an average twist of 156.48 ± 2.25° between two consecutive monomers, see above). Small variations in axial rise and twist between the monomers in the mature fibril result from the curvature of the protofilament, which is then stabilized by the interaction with a second protofilament.

**Structural comparison between plectasin wildtype and PPI42.** To elucidate the structural features that allow PPI42 to assemble into fibrils, while the plectasin wildtype did not fibrillate, we determined the crystal structure of the plectasin wildtype at a resolution of 1.1 Å (Supplementary Fig. 6). Our plectasin wildtype

structure shows a high similarity to the published crystal structure (PDBID: 3E7U[69]) with only minor differences in the N-terminal loop (all-atom RMSD: 0.316 Å). When we compared our wildtype plectasin crystal structure to the fibril cryo-EM structure of PPI42, we observed a significant structural difference in the N-terminal loop region between amino acids 9 and 14 (Fig. 4a). Two of these positions are mutated in PPI42 (D9S; Q14K). Therefore, we analyzed the coordination within this region in detail. We observed a network of polar interactions in the plectasin wildtype with (D9) forming a salt bridge to (N5). When mutated to (S9) in PPI42, the sidechain orientation changes and the sidechain forms a hydrogen bond to the (G6) backbone, which disrupts the network stabilizing this part of the N-terminal loop (Fig. 4b). Additionally, the (N5) sidechain is not internally coordinated, making the formation of polar interactions with neighbouring proteins possible, as was observed in the protein fibril. Notably, the sidechains of (E10), (D11), and (D12) form intermolecular interfaces both in the protein crystal of the wildtype (Supplementary Fig. 6) and in the fibril of PPI42. However, their orientation differs significantly between wildtype and PPI42 (Fig. 4c). This structural difference may contribute to the different arrangement with the mutant forming the fibril and the wildtype forming the crystal. The Q14K mutation additionally leads to a different coordination of (H18) (Fig. 4d), which plays an important role in forming the hydrophobic outer ring in the fibrils. In the plectasin wildtype, (H18) is coordinated in two different ways (50/50% distribution) stabilized by polar interaction with (Q14) and π-stacking with (F2), while in PPI42 only the π-stacking with (F2) occurs due to the mutation Q14K. The local electron density of (H18) in both structures is well defined (local resolution: 1.1 Å in plectasin wildtype crystal, 2.5 Å in PPI42 fibril).

Chemical shift perturbation studies of the backbone amide groups ($^1$H–$^{15}$N groups) at natural isotopic enrichment confirmed structural differences in (N5), (E10), and (D11) (Supplementary Fig. 7). We observed additional differences in the chemical shift of (C4), (K26), (G28), and (C30), indicating a difference in local structure and/or dynamics between PPI42 and wildtype (see Supplementary Discussion).

**Optical studies of fibrillation onset and mechanism.** As Rayleigh scattering is proportional to the sixth power of the particle radius, it readily captures small fractions of large oligomers. Hence, dynamic light scattering (DLS) was used to determine the onset and concentration dependence of fibril formation. The pH onset was measured by dilution to a final protein concentration of 2 mg/mL (450 μM), where the formation of the hydrogel was not so pronounced in the measured pH range. Based on the results, three size ranges for the DLS measurements were set, 0.1–10 nm to capture the monomeric fraction, 10–100 nm to capture medium-sized oligomers and 100–1000 nm to capture the fibrils.

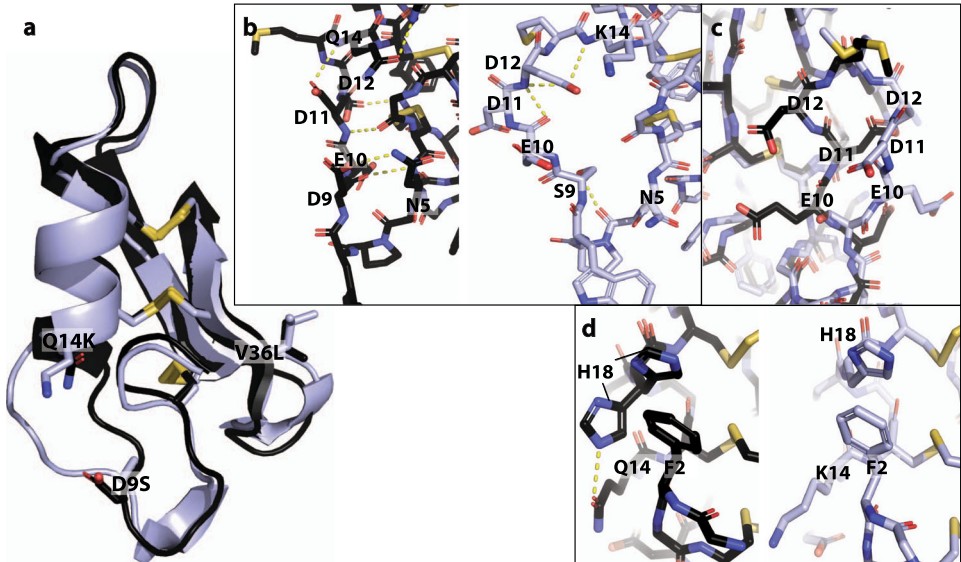

**Fig. 4 Comparison of the plectasin wildtype crystal structure (black) and PPI42 fibril structure (light blue). a** Overall comparison of the two structures shown as cartoon. The three mutated amino acids are highlighted as sticks. **b** Polar interactions in the N-terminal loop in the plectasin wildtype (left) and PPI42 (right). **c** Direct comparison between sidechain orientation of the acidic amino acids (E10), (D11) and (D12) in wildtype (black) and PPI42 (light blue). **d** Differences in the coordination of (H18) between the wildtype (left) and PPI42 (right). The wildtype showed two distinct orientations of (H18) while PPI42 only showed one orientation due to the Q14K mutation.

At pH 5, PPI42 showed no significant scattering from oligomers (Fig. 5a). From the cumulant analysis method of the autocorrelation function, a hydrodynamic radius $R_h$ of 1.53 nm was determined, which is close to the calculated monomeric $R_h$ of 1.37 nm[77], confirming that this fraction represented mainly monomeric species. Above pH 6, we observed a small fraction (<2.5% mass (Supplementary Fig. 5)) of medium-sized oligomers and fibrils. Therefore, we measured the concentration dependency of the formation of these large oligomers or aggregates at pH 6.5. We observed a concentration-dependent increase in the fraction of fibrils. The fraction of medium-sized oligomers did not increase with concentration indicating a direct addition of monomers to the fibril. Analysis of the mass percentage of each fraction showed a similar trend (Supplementary Fig. 5). This measurement indicated a pH and concentration-dependent equilibrium between monomer and protein in the fibril. The strong scattering influence of large polymers in DLS measurements implied that small variations in quantities of fibrils or other large particles can result in issues with exact quantification and reproducibility, which led to high standard deviations in our measurements.

**PPI42 fibril formation effect on function and structure**. The anti-microbial activity of PPI42 compared to plectasin wildtype was determined by measuring the minimal inhibitory concentration (MIC) against 112 Gram-positive bacteria including *Staphylococcus aureus*, drug-resistant strains such as methicillin-resistant *Staphylococcus aureus* (MRSA), *Staphylococcus epidermidis*, *Staphylococcus haemolyticus*, *Staphylococcus coagulase negative*, *Streptococcus pneumoniae*, *Streptococcus pyogenes*, *Streptococcus uberis*, *Streptococcus* Group A, B, C, G, *Streptococcus* non-hemolytic and clinical isolates (Fig. 5b). Plectasin wildtype and PPI42 appeared to be generally more potent against strains of *Streptococci* compared to strains of *Staphylococci*. PPI42 was more potent against strains of *Staphylococci* (up to eight-fold decrease in MIC) compared to plectasin wildtype while maintaining similar anti-microbial potency against *Streptococci*.

In order to investigate whether the fibril formation has an impact on the anti-microbial activity of PPI42, we performed a semi-quantitative radial diffusion assay (RDA). The antimicrobial activity of the plectasin wildtype and PPI42 was tested against *Staphylococcus carnosus* in different conditions as a function of protein concentration (Fig. 5c, Supplementary Fig. 5). PPI42 showed generally smaller clearing zones than the plectasin wildtype. However, despite confirmed gel formation of PPI42 at pH 5.5 and higher, no large differences in the size of the clearing zones could be observed between the tested conditions. This indicates a general hindrance in PPI42 diffusion in the plate compared to plectasin wildtype. The sample that was first dialyzed at pH 5.5, where gel formation was confirmed for PPI42, and subsequently dialyzed into pH 3.5, did not differ in its antimicrobial activity. We concluded, therefore, that undergoing fibril formation, does not compromise PPI42's anti-microbial activity.

To determine the structure-function relationship, we compared the accessibility of the proposed membrane and Lipid II binding site[68] and found that both binding sites are located in the protofilament centre (Supplementary Fig. 8), which makes it unlikely that PPI42 is active in its fibril state. Therefore, the fibrils in our RDA assay most likely dissolved during the time-course of the experiment.

We used circular dichroism (CD) spectroscopy to investigate whether PPI42 secondary structure was compromised upon fibril formation, and if the observed structural differences of PPI42 compared to the wildtype were only present in the fibril form. The far-UV CD spectra of wildtype and PPI42 were investigated as a function of pH (protein concentration of $c = 0.132$ mg/mL (30 µM)) (Fig. 5d). The spectra of PPI42 and plectasin wildtype were very similar, confirming high structural identity of wildtype and PPI42 in solution. At pH 7.5, the CD spectrum of PPI42 showed a clear change, but still resembled the presence of α-helix and antiparallel β-strands, indicating that the secondary structural elements remained intact. The observed change in the spectrum resulted most likely from a change in the N-terminal loop region as observed in cryo-EM, which affected the secondary structure. The observed change in CD spectrum was small in comparison to structural changes observed when amyloid fibrils are formed[78].

We investigated whether the fibril formation could be followed by ThT fluorescence measurements (Fig. 5e). Thioflavin T (ThT)

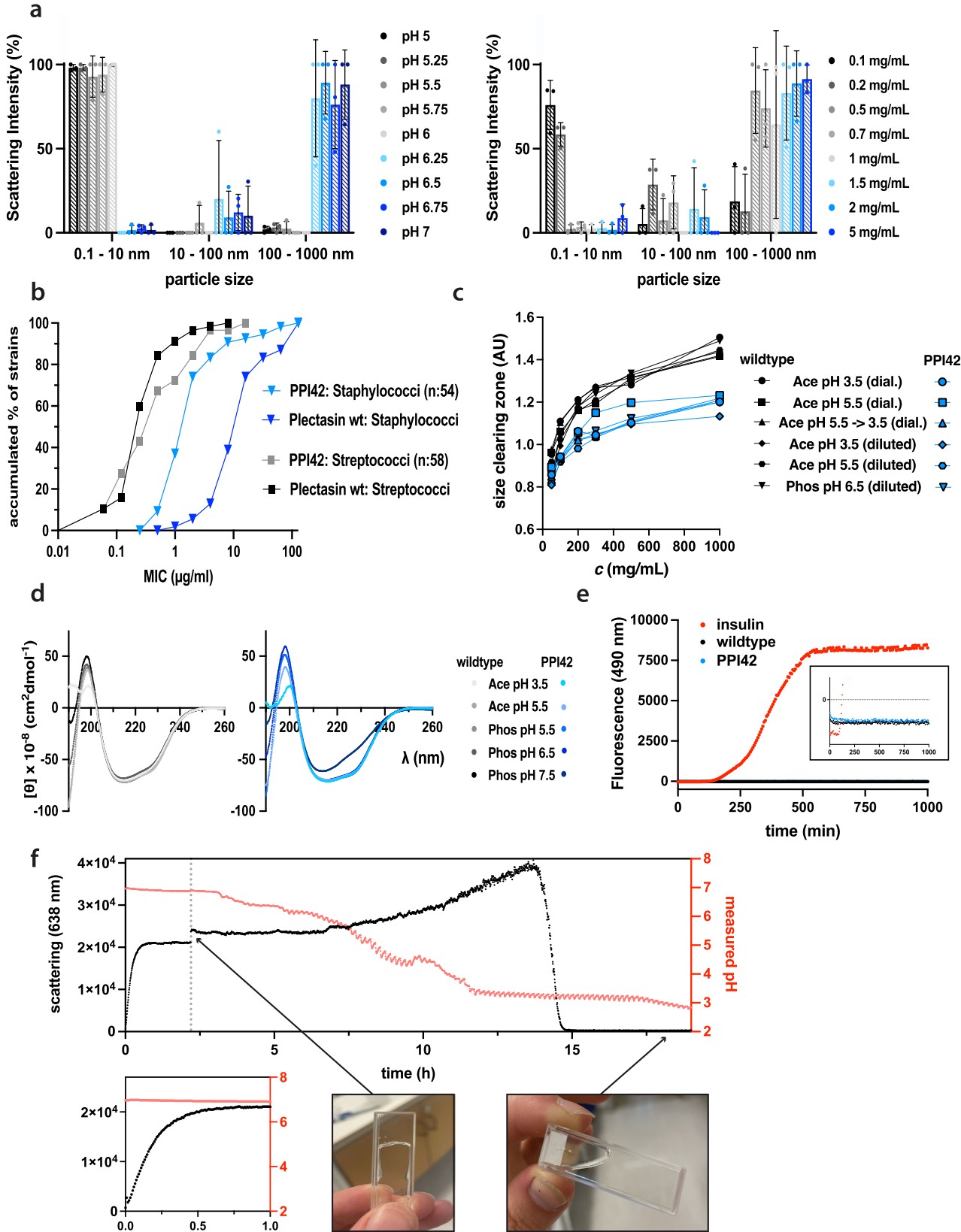

is widely used to follow amyloid fibril formation via fluorescence measurements[45] and has shown potential to follow other types of protein fibrils[64,65]. Insulin, which is known to form amyloid fibrils upon shaking at 40 °C[79], was used as a positive control. Insulin, plectasin wildtype and PPI42 were investigated at protein concentrations of 2 mg/mL. In contrast to insulin, neither the

wildtype sample nor PPI42 showed increased fluorescence. Since we observed protein fibrillation to be pH-responsive, we tested different buffer conditions, but no fluorescence increase could be observed, in contrast to an equilibrated insulin sample (Supplementary Table 2). Additionally, different temperatures and forms of mechanical stress were tested, but none led to an increase in

**Fig. 5 Molecular characteristics of the protein fibrils formed by PPI42. a** Dependence of fibril formation on pH ($c = 2$ mg/mL, left) and protein concentration (pH 6.5, right) determined by DLS. pH dependency was performed in phosphate buffer (protein concentration 2 mg/mL ($450\,\mu$M)). Concentration dependency was analyzed in phosphate buffer pH 6.5. Data are shown as %Intensity. Bars represent mean ± S.D. for 3 replicates (shown as dots). **b** Accumulated MICs of plectasin wildtype and PPI42 variant against 54 *Staphylococci* strains (blue) and 58 *Streptococci* strains (black). **c** Anti-microbial activity of the plectasin wildtype and PPI42 against *Staphylococcus carnosus* measured by semi-quantitative radial diffusion assay. The anti-microbial activity was measured by the size of the clearing zones as a function of protein concentration in different conditions relevant for this study. **d** Far-UV CD spectra of the plectasin PPI42 (blue, left) and wildtype (right, black) and as a function of pH ($c = 0.132$ mg/mL ($30\,\mu$M)). **e** Thioflavin T (ThT) fluorescence measurement over time on plectasin wildtype, PPI42 and insulin, which was used as a positive control ($c = 2$ mg/mL, $T = 40\,°$C). Insert shows ThT fluorescence monitored over time on wildtype and PPI42 in detail. **f** Light scattering and monitored pH of PPI42 diluted into phosphate buffer pH 7 and subsequently titrated with 1 M HCl. The grey line marks the starting point of the titration. Insert shows a zoom in between 0 and 1 h to visualize the increase in light scattering after dilution into pH 7, while the pH stays constant. Pictures show the sample at equilibrium after dilution into pH 7 (formed a gel) and after the titration with HCl (liquid). The monomeric state of PPI42 before dilution, at equilibrium at pH 7 and after titration was additionally assessed with NMR proving that the fibril formation was reversible (Supplementary Fig. 9). Source data are provided as a Source Data file.

fluorescence. These findings confirmed that PPI42 forms a type of protein fibrils that differ from amyloid fibrils.

We used pH titration in combination with light scattering to follow the fibril formation and investigate the reversibility of the process (Fig. 5f). Upon dilution into pH 7 ($c = 2$ mg/mL ($450\,\mu$M)), we observed an increase in scattering intensity following sigmoidal kinetics (Fig. 5f). At the equilibrium, gel formation was confirmed, and the sample was subsequently titrated with HCl, allowing for equilibration between each step. The pH was monitored during the whole measurement. We observed an initial increase in scattering intensity before it rapidly decreased to the level before fibril formation. NMR measurements showed full recovery of the monomer signal after titration (Supplementary Fig. 9), confirming the complete reversibility of PPI42 fibril formation. Plectasin wildtype meanwhile formed amorphous aggregates upon dilution to pH 7, and subsequently formed protein crystals after 15 h (Supplementary Fig. 9).

**NMR spectroscopy examination of factors affecting fibrillation.** We used real-time NMR spectroscopy to follow the kinetics of PPI42 fibrillation at various pH values at a protein concentration of 1.1 mg/mL ($250\,\mu$M). Due to their large size and long rotational correlation time, fibrils of PPI42 are not visible in liquid-state NMR, and the time series therefore followed the fraction of monomer over time (Fig. 6a and Supplementary Fig. 10). This allowed for an indirect way to follow the kinetics of protein fibrillation based on the integration of the isolated methyl groups of I22 and L36. Both, the fibrillation kinetics and the fraction of monomer in solution at equilibrium were pH-dependent. The pH-responsive fibrillation kinetics showed a transition between pH 6.0 and 7.0 (Fig. 6a), indicative of the titration of a histidine residue playing a role in fibrillation. The titration behaviour of chemical shifts for both protons in the imidazole rings of (H16) and (H18) was therefore determined by 2D $^1$H–$^1$H TOCSY NMR spectra (Fig. 6b). The $pK_a$ for (H16) was around 4.4, while the $pK_a$ for (H18) was around 6.4, which corresponds to the pH, where fibril formation was observed. (H18) and (H16) are involved in the hydrophobic interactions on the monomeric interfaces forming the outer hydrophobic ring, which might explain the pH-responsive fibril formation of PPI42 upon sidechain deprotonation (charge reduction). (S21) is involved in the interactions between the two fibrils and showed a similar pH response as (H18), indicating that the helix cap is structurally affected when fibrils are formed. For the plectasin wildtype no pH response of (S21) could be observed (Supplementary Fig. 10). The kinetic progress of fibrillation notably showed sigmoidal behaviour, strongly indicative of the need for an initial nucleation step to facilitate fibril growth. Consequently, fibrillation kinetics were investigated with regards to dependence on seeding and mono-mer concentration. To a sample of PPI42 freshly diluted into pH

6.75, $5\,\mu$L of a previously equilibrated sample in the same condition was added for seeding, which led to vastly accelerated fibrillation (Fig. 6c). This observation supported that a nucleation step occurs in fibril formation. Fibrillation kinetics as followed by NMR at varying protein concentrations at pH 6.75 finally showed that the supramolecular reaction had a strong dependence on protein concentration (Fig. 6d).

## Discussion

We have determined the structure, kinetics, and functionality associated with the pH-responsive self-assembly of the enhanced plectasin variant PPI42 into helical protein fibrils. The fibril structure characterized in this study is of a different kind than reported for other AMPs, containing α-helix and β-sheet structures of the native-like protein structure. Using a combination of AFM and negative stain EM, we characterized the network and surface morphology of the fibrils. The fibril network observed was consistent with negative stain EM and AFM measurements and the dimensions and elasticity were similar in different buffers, showing that the different buffer systems, ionic strengths (Supplementary Table 2), or treatments do not have any significant effect on the fibrils, which is considered advantageous for medical applications. The high-resolution structure of the protein fibrils formed by PPI42 solved at an overall resolution of 1.97 Å reveals that native-like monomers assemble into a right-handed helical superstructure consisting of two curved protofilaments. The fibril structure of PPI42 differs from the recently reported fibril structures of the anti-microbial LL-37[66] and uperin 3.5[25]. The structure is stabilized both, by polar as well as hydrophobic inter-subunit interactions. An outer hydrophobic ring may explain the structure of PPI42 fibrils. Polar interparticle interactions within the acidic amino acid patch (D9–D12) play a significant role in PPI42 fibrils as well as the crystal interactions of the plectasin wildtype. Within this patch, we observe significant conformational differences between the wildtype and PPI42 structures, whereas the remaining part of the protein is structurally very similar. This might explain the different types of self-assembly of wildtype and PPI42. A comparison between the cryo-EM structure of the mature fibril and of the isolated protofilament showed high similarity, indicating that protofilaments form prior to the supramolecular assembly into a curved mature fibril without additional structural rearrangement. The elasticity measured for PPI42 fibrils (0.7 and 1.1 GPa) was lower than typically observed for amyloid fibrils (2–4 GPa)[80,81], but is comparable to non-amyloid fibrils formed by ovalbumin[82].

The supramolecular association of PPI42 into fibrils is pH-responsive, which has been reported for synthetic and natural self-assembling proteins[4,46,48,83]. The equilibrium distribution of PPI42 between monomers and fibrils is dependent on pH and protein concentration. Increasing either of these two parameters

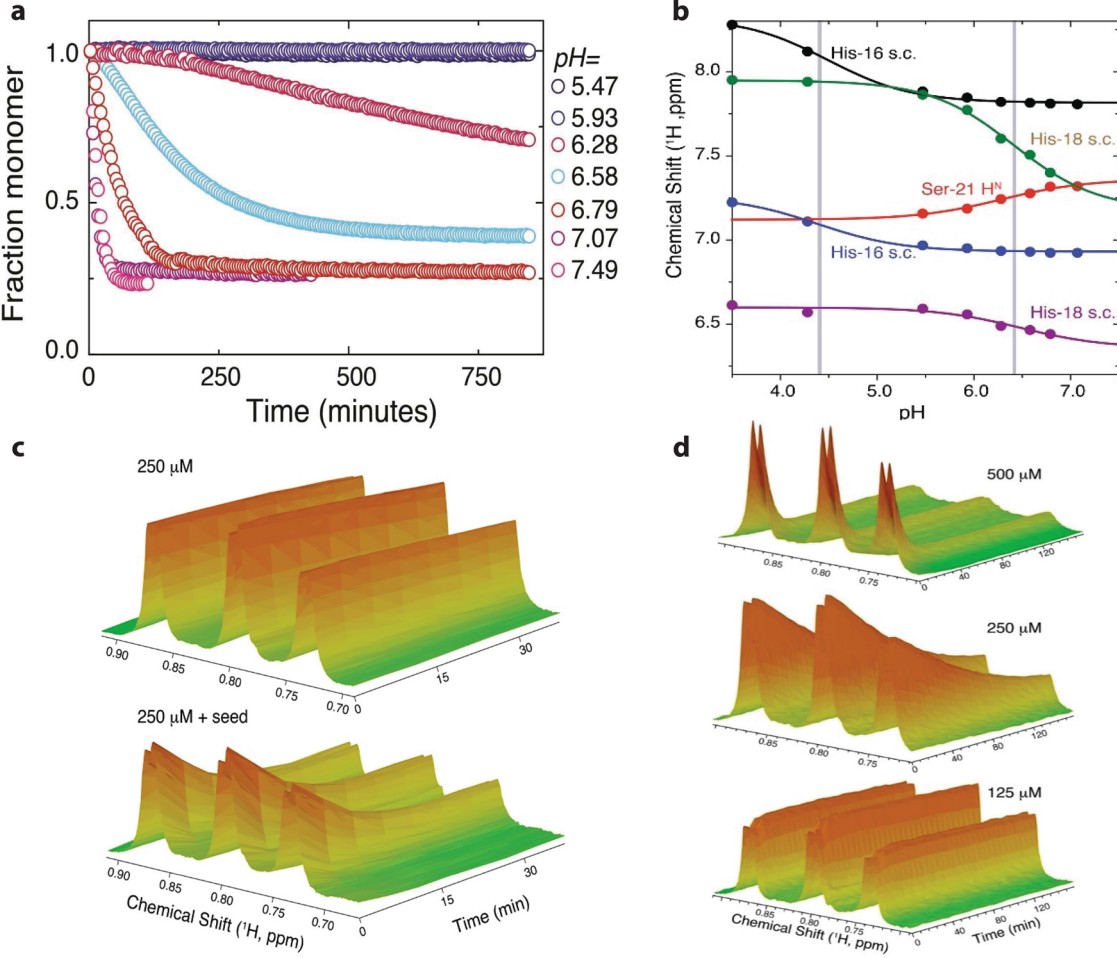

**Fig. 6 Fibrillation kinetics of PPI42 measured with NMR. a** The fraction of monomer was determined by integration of [1]H signals for the (I22) and (L36) methyl groups relative to time 0. **b** Titration curves of PPI42 (H16), (H18) (H[2] and H[4] in imidazole ring) and (S21) H[N] determined by identifying the signals with 2D [1]H–[1]H TOCSY. The grey bars indicate the $pK_a$ of the titratable sidechains. All measurements were conducted in 50 mM phosphate buffer. **c** Fibril formation kinetics with and without seeding with 5 µl of an equilibrated sample at pH 6.75. **d** Concentration dependence of fibrillation kinetics at pH 6.75. Source data are provided as a Source Data file.

shifts the equilibrium towards fibrils. Hence, a highly concentrated sample self-assembles at a lower pH than a lower concentrated sample, explaining the different pH onsets observed in this study. The small fraction of soluble, medium-sized oligomers does not show a clear trend with increasing pH or concentration. These results suggest that monomers are directly added to the fibrils end, without the formation of intermediate building blocks. The kinetics of fibril formation followed a sigmoidal curve, indicating nucleation dependent fibril formation. A seeding experiment with an already equilibrated sample showed accelerated kinetics, confirming the catalytic activity of the preformed fibrils, added as seeds. For the titratable sidechains of (H16) and (H18) and the backbone proton of (S21), $pK_a$ values were determined through pH titration. (H16) showed a $pK_a$ of 4.4, which is exceptionally low for an imidazole sidechain. We found that (H16) has on average a lower surface accessible area than (H18)[84], which has been shown to contribute to the variability of histidine $pK_a$ values[85]. (H18) sidechain chemical shifts (and the nearby S21) show a $pK_a$ of 6.4, which is very close to the pH where self-assembly becomes favourable. Thus, deprotonation resulting in a decrease in electrostatic repulsion of both histidine sidechains parallels the pH response of fibril formation for PPI42. This has been reported for the amyloid fibril formation of amylin[86] and α-synuclein[87]. Through the Q14K mutation the

coordination of (H18) in PPI42 changed, further supporting the crucial role of (H18) in the fibril formation. The reversibility of the fibril formation and the native-like protein structure within the fibril implied that only minor changes in the protein trigger its polymerization, consistent with recent reports that protein wildtypes are evolved on the edge of supramolecular self-assembly, which can be induced by a few mutations[88].

In order to investigate which methods are suitable to characterize the fibrils formed by PPI42 we applied a series of methods widely used for characterisation of amyloid fibril[8,78,89–91]. CD spectroscopy was applied to investigate whether PPI42 undergoes structural changes upon fibril formation. We indeed observed a change in the spectrum; however, compared to amyloid fibrils formed by AMPs[23,24,26], the general structural integrity of PPI42 remained intact. This is in agreement with the expected high structural identity of plectasin and the native-like structure of PPI42 in the protein fibril. Contrary to cross-α amyloid-like fibrils[25,64,65] and other fibril structures formed by AMPs[22–24,26–28], the fibrils of PPI42 proved to be negative for fluorescence detection with ThT.

PPI42 possess potent activity against a broad range of Gram-positive bacteria such as *Streptococci* and *Staphylococci*, including also drug-resistant strains such as methicillin-resistant *Staphylococcus aureus* (MRSA). Similar to the plectasin variant NZ2114,

PPI42 shows enhanced activity against *Staphylococci* compared to the wildtype[72]. Both variants have a mutation at D9 reducing the protein's charge at physiological pH (PPI42: D9S; NZ2114: D9N). As D9 is involved in the membrane binding of plectasin[68], the increased activity of both variants against *Staphylococci* might result from enhanced binding to the negatively charged membrane. Based on the position of the binding site for the bacterial membrane and lipid II, essential for the activity of plectasin, we believe that the fibrillar form of PPI42 is not active. Consequently, the fibril would serve mainly as a deposit for the protein. We show that the equilibrium between fibrillar and monomeric forms depends on the protein concentration. As such, the fibrillar form might therefore be used for slow release over time. The use of natural occurring self-assembly has inspired several functional biological materials that consist of self-assembling units[41]. Self-assembly has also been suggested to provide protection against proteases over time[68], which would need to be proven for the PPI42 fibrils. It has been suggested that tailoring the self-assembly might allow targeted anti-microbial therapy[92]. pH as an external trigger for self-assembly has already been used for the self-assembly of AMPs[46] and through introduced mutations the pH responsiveness could be tuned[48]. Our study lays the foundation for the characterization of a different fibril type formed by anti-microbial peptides (AMPs) and might inspire the design of novel AMPs.

## Methods

**Sample preparation**. If not stated otherwise, all plectasin samples were dialyzed into the desired condition using Slide-A-Lyzer™ 2000 MWCO dialysis cassettes (Thermo Fisher) with a buffer exchange after 2 and 4 h and then continued overnight, ensuring a dilution of at least 200 times in each step. The concentration of plectasin wildtype stock solution was $c = 39$ mg/mL (8.9 mM) and of PPI42 $c = 37$ mg/mL (8.4 mM). If not gelled, the protein concentration after dialysis was measured using a NanoDrop™ 8000 Spectrophotometer. The pH reported was the measured pH of the dialysis buffer, if not stated otherwise. Different buffers were chosen to ensure sufficient buffer capacity at different pH values. For the lower pH range (pH 3.5–5.5) sodium acetate buffer was chosen, if not stated otherwise. For the neutral pH range (pH 5–7.5) phosphate buffer was chosen, if not stated otherwise.

### Anti-microbial activity assays

*Minimal inhibitory concentration (MIC) assay*. MICs were determined by the microdilution broth method according to National Committee for Clinical Laboratory Standards/ Clinical and Laboratory Standards Institute (NCCLS/CLSI) guidelines and as described by ref. [67]. Freshly prepared overnight colonies were suspended (turbidity of 0.5 McFarland units) and further diluted in Mueller-Hinton BBLII medium (Becton Dickinson). The medium was supplemented with 2–5% defibrinated horse blood for all *Streptococcus* species. The bacterial suspensions were added to wells containing two-fold peptide dilutions. The polypropylene trays (Nunc) were incubated at 35 °C in ambient air for 16–20 h for *Staphylococcus* spp. and 24 h for *Streptococcus* spp.

*Radial diffusion assay*. Preparation of target strain, Staphylococcus carnosus: Stock of target strain was incubated at 37 °C overnight in anaerobic conditions (AnaeroGen 2.5 L, Thermo Scientific) on to LB agar plate then resuspended in 0.9% NaCl, 20% glycerol to a McFarland turbidity standard of 1 (OD$_{600}$ ~0.870). Aliquots of 150 μl are frozen at −80 °C. The colony-forming unit per mL (CFU/mL) was determined. The RDA media (5.5 g Mueller-Hinton II Broth, Beckton Dickinson; 7.5 g Agarose High resolution, Sigma–Aldrich, 500 ml MilliQ water) was autoclaved (121 °C; 15 min) and cooled down to ~42 °C. S. carnosus was added 30 ml medium to a final CFU/mL of 5x10e$^5$. Plates were prepared using Omnitray with NUNC-TSP lid (both Thermo Scientific) and kept at 4 °C for at least 20 min to solidify. The lid was discarded, and the wells were loaded with 10 μL sample. The RDA plate was incubated at 37 °C overnight. The plate was coloured with 1.5 mM Thiazolyl Blue Tetrazolium Bromide (MTT) (Sigma–Aldrich).

**Atomic force microscopy**. Atomic force microscopy was performed on PPI42 stock solution dialyzed against 10 mM acetate buffer pH 5, 10 mM citrate buffer pH 5, H$_2$O pH 5, 10 mM histidine buffer pH 6.5. Different buffer systems and pH values were chosen to ensure the fibril structure was not affected by the different buffers necessary for this study. High protein concentrations (~20 mg/mL (4.5 mM) after dialysis) were chosen to ensure gel formation also at a slightly acidic pH. All samples formed a gel. A piece of gel was gently transferred to a silicon

wafer and dried at ambient conditions. All samples were washed three times with MilliQ water and dried again. A Multi-mode 8 AFM (Bruker Nano) with Peak-Force Quantitative Nanomechanical Mapping (QNM) mode was used for imaging with simultaneous mapping of topography and elasticity. TAP150A probes (Budget Sensors) with a nominal spring constant of 5 N/m were used. For determining the elasticity of the samples, the following calibration procedure of the probe was used: (1) deflection sensitivity calibration on sapphire; (2) spring constant calibration using thermal tuning; (3) tip radius calibration using polystyrene test sample (Bruker QNM sample kit, PS film, Nominal elastic modulus 2.7 GPa); (4) calculation of elastic modulus using DMT model. The same probe was used for all measurements.

**Circular dichroism**. Plectasin stock solutions were dialyzed into 10 mM acetate buffer pH 3.5 and pH 5.5 and filtered if no gel had formed (0.02 μm). CD requires only low protein concentrations, to not saturate the detector. Acetate and phosphate buffer were chosen due to the least interference with the method. Samples of 30 μM (0.132 mg/mL) were prepared by dilution with the filtered dialysis buffer. The CD spectrum was measured with a JASCO J-1000 spectrometer from 190 to 260 nm with 1 mm optical path, using Spectra Manager CFR for instrument control and analysis. Five spectra were accumulated for each measurement. Ellipticity $\theta$ (mdeg.) was converted to molar ellipticity $[\theta]$:

$$[\theta] = \frac{100 * \theta}{c[M] * l(\text{cm})}$$

$c$ is the protein concentration and $l$ is the pathlength.

**Cryo-EM grid's preparation**. PPI42 stock solution was diluted with 10 mM acetate buffer pH 5.5 to a final protein concentration of $c = 10$ mg/mL and dialyzed against 10 mM acetate pH 5.5 as described above. Dilution prior to dialysis was observed to result in a more homogenous sample and good coverage of the whole grid. 3.5 μL of the sample was applied to glow-discharged 2/2 Quantifoil holey carbon grids (Quantifoil Micro Tools GmbH, Germany) and plunge frozen in liquid ethane with a Vitrobot Mark IV (Thermo Fisher Scientific) (6 s blot time, blot force 0, drain time of 0.5 s). The sample was observed at the beamline CM01 of the ESRF (Grenoble, France)[93] with a Titan Krios G3 (Thermo Fischer Scientific) at 300 kV equipped with an energy filter (Bioquantum LS/967, Gatan Inc) (slit width of 20 eV). 8443 images were recorded automatically on a K2 summit direct detector (Gatan Inc.) in counting mode with EPU (Thermo Fisher Scientific). Movies were acquired for a total exposure of 4 s and 100 ms per frame resulting in 40 frame movies with a total dose of 46.8 e$^-$/Å$^2$. The magnification was ×165,000 (0.827 Å/pixel at the camera level). The defocus of the images varies between −0.5 and −1.5 μm.

**Cryo-EM image analysis of the mature fibril**. Image processing was performed in RELION 3.1[94] Movies were drift-corrected using MotionCor2[95]. CTF estimation of the micrographs was performed using GCTF[96]. Empty field of view and crystalline ice micrographs and micrographs with an estimated resolution lower than 4.5 Å (CTF estimation step) were removed, resulting in a set of 7241 micrographs. An initial set of particles obtained by manual picking and initial 2D class averages were used to pick automatically all micrographs using a rather conservative threshold. Improved 2D class averages for the superstructure clearly displayed an axial 2-fold symmetry. An axial rise $z = 24.8$ Å was measured on their Fourier transforms. An initial 3D reference was obtained by searching iteratively for the best match between 2D projections of a 3D reconstruction computed with some given helical parameters and an ab initio 2D class average[97,98]. This method identified clearly the axial rise $z = 24.9$ Å but failed to determine reliable azimuthal angles phi. A 3D reconstruction with $z = 25.1$ Å and phi = 16.1° was arbitrarily chosen as an initial model for 3D refinement. The latter was further low pass filtered to 40 Å to avoid any model bias. A first 3D refinement with two-fold symmetry resulted in a 3.35 Å resolution map. After two iterations of a 3D classification (without alignment) followed by a 3D refinement, a 3 Å map was calculated. The plectasin monomer was fitted unambiguously. Using the known axial rise ($z = 25.1$ Å), an initial azimuthal angle phi = 15.6° could be determined from the position of two successive plectasin monomers docked in the map. A second automatic picking was performed with a lower threshold to generate more particles (2,313,914 in total). The 2D classification showed a vast majority of the superstructure (1,750,332 particles) as well as a small population of single fibril (99,784 particles—see below). A first 3D refinement with two-fold and helical symmetry followed by a 3D classification and particle polishing resulted in a 2.5 Å resolution map calculated from 764,822 particles. CTF refinement in RELION was repeated two times which improved the resolution of the reconstruction to 2.05 Å resolution. A final CTF refinement (beam tilt, trefoil and 4th order aberrations only) was performed with 10,000 particles per optic group to account for the variations in coma alignment during the length of the data collection. After the final 3D refinement, the final 3D map for the superstructure was calculated (final helical parameters $z = 25.10$ Å and phi = 15.75°) and a resolution of 1.97 Å was determined by Fourier Shell Correlation (FSC) at 0.143. The asymmetric unit was composed of seven monomers arranged in a near-helical way with an average axial rise of 3.75 ± 0.15 Å and an average twist of 156.48 ± 2.25° between them.

**Cryo-EM image analysis of the isolated protofilament**. From the 2D classification of the superstructure, a small proportion of single fibril was identified. A separate round of 2D classification allowed the isolation of a homogenous group of single fibrils (66 272 particles). An axial rise $z = 3.78$ Å was measured from the Fourier transform consistently with the average axial rise ($3.75 \pm 0.15$ Å) found between monomers of the mature fibril. The 3D reconstruction of the single fibril was obtained after a 3D refinement using as a starting model a cylinder of 70 Å in diameter and using as initial helical parameters the average one found between monomers of the mature fibril ($z = 3.75$ Å and phi = 156.5°, see above). This resulted in a 3D map at the resolution of 3.35 Å as determined by Fourier Shell Correlation (FSC) at 0.143. The final helical parameters were determined as $z = 3.76$ Å and phi = 156.5°.

**Cryo-EM model refinement**. The crystal structure of the plectasin wildtype was the first rigid-body fitted inside the cryo-EM density maps in CHIMERA[99]. The atomic coordinates were then adjusted manually in COOT[100] and refined in the cryo-EM map with ROSETTA[101] and PHENIX[102]. The refined atomic models were visually checked and validated with MOLPROBITY[103]. The cryo-EM related figures were prepared with CHIMERA and CHIMERAX[104]. The data collection and the model statistics are summarized in Supplementary Table 4.

**Dynamic light scattering**. Plectasin stock solutions were dialyzed against 10 mM acetate buffer pH 4.5 as described above, filtered (0.02 μm) and the concentration was measured. A protein stock solution of 20 mg/mL (4.5 mM) was obtained by dilution with the filtered dialysis buffer (0.02 μm). The respective formulations were obtained by a 10 times dilution in phosphate buffer with the desired pH. We found a final concentration of 2 mg/mL (450 μM) to give a good signal to noise ratio in DLS. The measurement was performed with a DynaPro® Plate Reader$^{TM}$ II (Wyatt Technology) using Aurora 384 LV/EB plates (Brookes Life Science Systems). All measurements were performed at 25 °C with 5 s acquisition time and 20 acquisitions per well. All formulations were measured in triplicates. The analysis was performed DYNAMICS and final graphs were made with Origin® 2019(OriginLabs) and GraphPad Prism.

**Fibre diffraction**. PPI42 stock solution was dialyzed against 10 mM acetate buffer pH 5.5. The gelled sample was dried for 48 h and was analyzed using a Supernova CCD diffractometer from Agilent.

**Gel formation assay**. Plectasin stock solutions was diluted into 50 mM phosphate buffer pH 7 to final concentrations of 0.5, 1, and 5 mg/mL (115 μM, 230 μM, and 1.15 mM). An air bubble was gently introduced below the surface and the tube was turned. Each sample was assessed on the stiffness of the gel based on the ability to stay at the top of the tube and whether the bubble raised to the top.

**Negative staining**. PPI42 stock solution was dialyzed against 10 mM acetate buffer pH 5.5. To ensure comparability with AFM measurements, the same protein concentration was chosen. The samples were absorbed to the clean side of a carbon film on a carbon–mica interface and stained with 2% sodium silico-tungstate (pH 7.4). The carbon was transferred to a 400-mesh copper grid. Images were taken under low dose conditions ($<30$ e$^-$/Å$^2$) with defocus values between $-1.2$ and $-2.5$ μm on a Tecnai 12 LaB6 electron microscope at 120 kV accelerating voltage using CCD Camera Gatan Orius 1000.

**NMR measurements**. Plectasin stock solutions were dialyzed against 10 mM acetate buffer pH 3.5. Samples of 500 μl volume protein solution were prepared by dilution into the final buffer. The samples contained 250 μM concentration (unless indicated otherwise), which we found gave a good signal to noise ratio for our measurements, in 50 mM phosphate or 50 mM acetate buffer containing 10% v/v D$_2$O as the lock substance. Buffers were chosen due to the least interference with the relevant signals in the spectra. All NMR spectra were recorded on an 800 MHz Bruker Avance III NMR spectrometer equipped with an 18.7 T Oxford Magnet and a Bruker TCI CryoProbe at 298 K. Protein aggregation over time was followed by a sequence of one-dimensional $^1$H NMR experiments employing excitation sculpting as the water suppression scheme. The series of one-dimensional $^1$H NMR was implemented as a pseudo-2D experiment, which sampled 16,384 complex data points during an acquisition time of 1.27 seconds and accumulated 128 transients with an inter-scan relaxation delay of 1 second per time point.

$^1$H–$^{15}$N HSQC experiments were acquired at pH 4.5 for the wildtype and PPI42 on a Bruker Avance III instrument equipped with an Ascend 800 magnet by sampling the FID for 160 and 20 ms in the direct and indirect dimension, respectively.

2D $^1$H–$^1$H TOCSY NMR spectra were acquired by sampling the FID for 117 and 14.5 milliseconds by acquiring $1024 \times 128$ complex data points in the direct and indirect dimensions, respectively. For these $^1$H–$^1$H TOCSY NMR experiments, 16 transients were acquired with an inter-scan relaxation delay of 1 second. All NMR spectra were processed with ample zero filling in all spectral dimensions in Bruker Topspin 3.5 pl7 software and integrated into the same software. Data were plotted in proFit 7 (Quantum Soft). Titration curves were fitted to the Henderson-

Hasselbalch equation in proFit 7 as

$$\delta_{obs} = (\delta_A + 10^{pKa-pH} * \delta_{HA})/(1 + 10^{pKa-pH}),$$

where $\delta_{obs}$ is the measured chemical shift, $\delta_A$ is the chemical shift of the deprotonated form and $\delta_{HA}$ is the chemical shift of the protonated form. The pH of the sample was determined after the measurement.

**Reversibility of fibril formation**. Plectasin stock solutions were dialyzed into 10 mM acetate buffer pH 3.5. Samples of 500 μL were prepared in a quartz cuvette by dilution into 50 mM phosphate buffer pH 7 to a final protein concentration of 1.1 mg/mL (250 μM). We found this concentration to give a sufficient signal to noise ratio without saturating the detector. Measurements were started immediately. The fibril formation or aggregation was monitored by simultaneous static light scattering and pH measurement using a Probe Drum instrument (Probation Labs AB, Sweden). Static light scattering was measured with a laser of 638 nm at 90° angle. Acid titration (only for PPI42) was performed by injection of 0.5 μL 1 M HCl allowing for equilibration of 10 min between each injection.

**Thioflavin T fluorescence measurements**. Plectasin stock solutions were dialyzed against 10 mM acetate buffer pH 5.5. A final concentration of 2 mg/mL (450 μM) was obtained by dilution with the dialysis buffer and ThT was added to a final concentration of 20 μM. The fluorescence was monitored after each cycle at $490 \pm 10$ nm with excitation at $440 \pm 10$ nm with a gain of 1000 using FLOUstar Omega multi-mode microplate reader (BMG Labtech). Omega software was used to control the plate reader and export the data. Insulin was fibrillated at 40 °C with agitation (600 rpm) for 150 s every 5 min[79]. Additionally, the fluorescence of a sample containing insulin fibrils at a protein concentration of 2 mg/mL was compared to freshly dialyzed plectasin in 10 mM acetate buffer pH 3.5 and pH 5.5 and a sample diluted into 10 mM acetate buffer pH 5.5 and 10 mM phosphate buffer pH 6.5, equilibrated for 12 h was compared (end-point measurement).

**X-ray structure determination**

*Crystallization*. Crystallization conditions of plectasin were optimized from a sparse matrix screen[71] by Molecular Dimensions. Crystallization conditions were optimized to: 2 μL of 20 mg/mL plectasin wildtype in 10 mM Acetate pH 5.5 added to 2 μL reservoir of 0.1 M NH$_4$Ac, 0.1 M Tris pH 8.5 and 40% isopropanol. Crystals formed overnight.

*Data collection and structure determination*. Crystals were flash cooled directly in liquid nitrogen. X-ray data were collected at 100 K at the BioMAX beamline at MAXIV, Lund, Sweden[105]. Data reduction was performed with the autoPROC toolbox[106] using XDS/XSCALE[107] and Pointless[108].

*X-ray structure determination*. The crystal structure was determined using the CCP4 program suite[109] with molecular replacement using Molrep[110]. The previously solved crystal structure of plectasin 3E7U[69] was used as a template. Restrained positional and anisotropic B-factor refinement was performed in REFMAC5[111]. The hydrogen atoms were included in riding positions. Data collection and refinement statistics are given in Supplementary Table 5.

**Reporting summary**. Further information on research design is available in the Nature Research Reporting Summary linked to this article.

## Data availability

The authors declare that the data supporting the findings of this study are available within this publication. Source data are provided with this paper for Figs. 5a-f, 6a,b, S5b and S9b. Additional raw data are available from the corresponding author upon reasonable request. The atomic structures determined in this study are deposited in the PDB database and the EM data bank. PDB identifier for mature fibril is 7OAE, for the isolated protofilament 7OAG, for the plectasin wildtype crystal structure 7O76. EMBD identifier for mature fibril is EMD-12775, for the isolated protofilament EMBD-12776.

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

## Acknowledgements

This work was funded by European Union's Horizon 2020 research and innovation program (grant agreement no. 675074) (P.H., A.N., W.S. and G.H.P.). We thank Birgitte Andersen and Ida Ahlmann Ellingsgaard (Novozymes A/S) for their input on previous plectasin studies, supply of purified material and their help on the activity measurements of plectasin. We thank Rahmi K. Elfa for her work on the crystallization of the plectasin wildtype. We thank Ingemar André (Lund University) for useful discussions about surface accessible areas in protein fibrils. We thank Thom Leiding and Mattias Törnquist (Probationlabs) for their input on the setup of simultaneous pH and light scattering measurements. We acknowledge the provision of in-house experimental time from the CM01 facility at the ESRF. This work used the EM facilities at the Grenoble Instruct-ERIC Center (ISBG; UMS 3518 CNRS CEA-UGA-EMBL) (E.K. and C.M.-D.) with support from the French Infrastructure for Integrated Structural Biology (FRISBI; ANR-10-INSB-05-02) and GRAL (G.E. and G.S.), a project of the University Grenoble Alpes graduate school (Ecoles Universitaires de Recherche) CBH-EUR-GS (ANR-17-EURE-0003) within the Grenoble Partnership for Structural Biology. The IBS Electron Microscope facility is supported by the Auvergne Rhône-Alpes Region, the Fonds Feder, the Fondation pour la Recherche Médicale and GIS-IBiSA. We acknowledge MAX IV Laboratory for time on BioMAX under Proposal [20190334] (P.H.). Research conducted at MAX IV, a Swedish national user facility, is supported by the Swedish Research council under contract 2018-07152, the Swedish Governmental Agency for Innovation Systems under contract 2018-04969, and Formas under contract 2019-02496. NMR spectra were recorded using the 800 MHz spectrometer at the NMR Center DTU, supported by the Villum Foundation (S.M.).

## Author contributions

C.P., A.N. and P.H. designed the study. C.P. conducted, analyzed and interpreted DLS, CD, fiber diffraction and ThT experiments and RDA assay. G.E. processed cryo-EM data, and built atomic models, E.K. conducted cryo-EM experiments. G.S. and C.M.-D. assisted in cryo-EM experiments, model building and structural analysis. S.M. conducted, analyzed and interpreted NMR experiments. G.Z. conducted, analyzed and interpreted AFM experiments. D.R.S. coordinated and interpreted in vitro and in vivo activity data. P.H.M. and D.S. conducted in vitro activity assays. L.A.N. conducted in vivo activity and

toxicology assays. P.H. conducted, analyzed and interpreted X-ray structure determination. C.P. wrote the manuscript with support from A.N. and P.H. P.H., A.N., W.S. and G.H.J.P. supervised the study. All authors corrected and approved the final manuscript.

## Funding

## Competing interests

The authors declare no competing interests.
