## [Peer Review File · Nature Communications]

REVIEWER COMMENTS

Reviewer #1 (Remarks to the Author):

Pohl et al described the first (to the best of my knowledge) CryoEM structure of a fibril-forming AMP.

The structure described is of PPI42- plectasin variant D9S Q14K V36L, which forms fibrils more readily than the wt. Fibrillation is reversible in acidic conditions.

The structure is an assembly of the folded monomers, which is different from amyloid fibrils, and resembles actin fibrils in the aspect. DLS measurements indicated a pH and concentration-dependent equilibrium between monomers and fibrils. Direct monomer addition to the fibril is expected in a folded monomer assembly. Antimicrobial activity showed a slight reduced activity of the mutant vs the wt, but it is unclear if the effect is related to fibrillation or to the mutations irrespectively. It is hard to control fibrillation conditions to answer this question. Histidine residues seems to serve as the control switch for pH-dependent fibrillation.

Conceptual comments:

1. Since only the variant but not the wt plectasin form fibrils, it is hard to project the relevancy of this fibril formation in nature. Nevertheless, it can still promote the design of AMP supramolecular structures that provide stability with improved clinical and technological usage.
2. Another aspect is lacking to indicate the functional relevancy of the self-assembly. Is the fibril only acting as a reservoir or the fibrils themselves have direct activity? This would not be a trivial assessment, and I am aware of the broad scope of experiments needed to answer this question.

Technical comment:

1. Intro: "Protein fibrils have been found to serve as structural elements in long-term memory in microorganisms^{1,2}." – I don't think that the memory is in the microbes (ref1 is SAA – a human protein ;ref 2 is Aplysia CPEB).
2. I am confused about the variant: in the introduction it was mentioned: "Previously, we could show that the introduced mutations, which were originally implemented to increase anti-microbial activity, were predicted by MD simulations to only have a minor effect on the overall structure of plectasin⁴² (Figure S1)". But the antibacterial activity shown here suggest reduced activity. This should probably be better addressed in the intro. Can it be related to the type of assay conducted? (see my comment #12 below).
3. Nomenclature is difficult- the two fibrils in the supramolecular structures are sometimes called fibrils and sometimes strands, which is especially confusing with b-strands. The convention, I think, is protofilament or protofibril, which assemble into "fibers" or "mature fibrils/fibers"
4. Structure Table – I would add Molprobity and ClashScore percentiles.
5. Figure 2- explain the "N" and "I" in the figure legend. The figure is unclear regarding the arrangement of the individual monomers. Maybe color each monomer in a different color?
6. Figure S4 panel – it would be helpful to indicated on panel H the residues constituting the core (those indicated in panel F) for orientation.
7. Figure 3 - The right and left panels show different clusters (and cluster 1 is not hydrophobic entirely, as it contains histidine) I suggest using two colouring schemes for the two clusters and trying to also show a joint view of the clusters in CPK. I am not sure, but maybe it will be easier if the structure would be shown as ribbons of secondary structure with the clusters in CPK. Also, the 90deg arrow is not indicative of the rotation. I am confused about panel E right: is panel B also showing more stabilizing interactions between the fibrils? Maybe a better coloring scheme will help with orientation here to

distinguish between monomers and between protofibrils?

8. The interface between the two protofibrils seems weak/small according to figure 3E. Was the solvent accessible surface area between monomers and between protofibrils calculated? It is often an indication of the compactness and relevancy of interfaces. It is directly correlated with free energy of binding.

9. Figure 4- the structural differences can be related to the mutations but also the methods (crystal vs solution), as mentioned. Impossible to tell for sure. The two coordination of H18 in panel D should be mentioned in the legend. In the CryoEM, the single rotamer observed can be an issue of a lower resolution. The crystal structure of the mutant was not obtainable? If so, for what reason?

10. A view of the secondary structure is really missing for me to see that discussed interactions between protofibrils/monomers, and the overall packing. The one given in Figure S5 is not sufficient as it doesn't include the specific interactions discussed.

11. Figure S5 – I am lost in this figure. The two protofibrils in the superstructure do not look symmetric/equivalent. Is that possible in the asymmetric unit? what is the superimposition in panel B?

12. Antibacterial activity – it is plausible that in a case of a fibril-forming AMP, it is more indicative to conduct the assay in solution and not by diffusion.

13. Figure S6 – I don't understand the difference in the DLS results to the ones shown in figure 5.

14. In my opinion, the indicated resolution is misleading, and in the case of CryoEM, it is important to indicate the range of resolution in the main text. Looking at the map fitting, it is clear to me that some areas are poorly fitted, which is fine, and acceptable, but the indication of salt bridges and other tight interactions, for example, should be mentioned with regard to the resolution of the local region.

Reviewer #2 (Remarks to the Author):

In their communication, "Reversible supramolecular assembly of the anti-microbial peptide (AMP) plectasin into helical non-amyloid fibrils" Christin Pohl et al. have studied the non-amyloid fibril formation by a mutant variant of the antimicrobial peptide plectasin-providing high-resolution structure using Cryo-EM.

- What are the noteworthy results?

The results are interesting and worth discussing in context of the structural insight obtained from Cryo-EM. However, to this reviewer it is unclear as to what is the background of the study. Lots of lacunae lies in terms of the functional implication of the entire structure elucidation. The introduction is elaborate, yet fails to lay the foundation for the study, the peptide or its variant. The major aim is not defined or established anywhere along the manuscript. Non-specific readers have to go back and forth with their previous manuscript in order to gain some understanding of the peptide or its variant.

- Will the work be of significance to the field and related fields? How does it compare to the established literature? If the work is not original, please provide relevant references.

Work is original and the detailed structural information as obtained from Cryo-EM is noteworthy, as it is a gradually expanding field in structure determination.

However, to this reviewer it seems important to correlate the structures to the peptide's functional attributes that, in turn, should enhance the readability and application of such structural studies. If not at the interface of peptide function, at least an elaboration on the immediate application of such structural information is necessary, that appears to be

largely missing from this study.

- Does the work support the conclusions and claims, or is additional evidence needed?

The work introduces AMPs and amyloids and goes on to define the structure of a mutant variant of plectasin (without any direct introduction to this mutant, its function or plectasin as such) and using several biophysical methodologies proves that the pH-responsive self-assembly of PPI42 results in forming non-amyloid protein fibrils.

The authors suggest "supramolecular fibrils of plectasin may have the potential to provide a slow release of the monomer, which is known to be bioactive." Given their study, this claim can only be a hypothesis and further experimentation would be necessary to claim this as an immediate application of their study (suggested in their discussions section). Moreover, the authors themselves have claimed that the fibrillar conformers have no effect on the functional potency of the peptide. Thus, it is difficult to gauge as to what is the claim that the authors want to put forward in their study.

- Are there any flaws in the data analysis, interpretation and conclusions? - Do these prohibit publication or require revision?

While the structure determination in itself looks good enough, the study in whole seems to lack a defined implication altogether. Some of the major factors that are quite off-putting are as follows:

The study suggests that the peptide undergoes fibrillation in different pH conditions and peptide concentrations. This is already a known and established fact for several proteins and peptide systems. In fact high-resolution dynamic studies are also available to correlate the aggregation propensities of peptides. While several of these studies have indeed taken "amyloids" into consideration, the approach is relevant even for "non-amyloids." Why is "non-amyloid" conformation of the peptide reinforced throughout the length of the manuscript, remains unclear, at least to this reviewer as to what does it signify?

"Plectasin variant PPI42 formed a gel at peptide concentrations of approx. 5 mg/mL and higher, during dialysis at pH 5 and above" this too has been seen in several biologically active peptide systems. But the authors do not provide any introduction to this variant of plectasin and its antimicrobial attribute that might help in giving any understanding of the pH dependence of only the variant and not the wildtype.

The authors claim that the gel formation is irrespective of the chemical nature of the buffer system. However, it is very difficult to gain a clear picture as they have defined the use of different peptide concentrations and different buffer conditions for the different biophysical studies. It has been extensively shown in literature that a protein's folding or fibrillation attribute is largely influenced by the pH and sample concentrations. Especially, with more and more literature being made available about protein's liquid-liquid phase separation and gel formation that it is essential to define why a particular pH and peptide concentration has been used for a particular study and what the physiological implications are. For an antimicrobial peptide (AMP), a peptide concentration of 5mg/ml appears to be too high, thus lowering the implication of the structural insight into the fibrillar forms. Is this concentration even relevant to the peptides function? What is its effective microbial inhibitory value? Has the authors performed any experiment to determine the same? It seems only logical that the authors provide atleast some data on the cytotoxicity at the concentrations used for the fibrillar structure determination. Although it is known from literature that protein fibrils are less cytotoxic as opposed to the intermediate oligomers; however, this is not always true, since fibrils often interfere with the homeostatic conditions. Therefore, even the clinical implications suggested in the discussion has to be tested at the used concentrations against the host cellular systems, while providing antimicrobial data for the same.

The study suggests "significant structural difference in the N-terminal loop region between

amino acids 9 and 14". It is possible to find a direct correlation between this flexible loop regions to the pH sensitivity of the distinct fibrillar pathway for the variant using heteromolecular 2D NMR. More so, applying experiments to gain atomic resolution into the dynamics of the early folding events might be helpful in gaining the differential aggregation pathway, particularly, in the different pH systems.

- Is the methodology sound? Does the work meet the expected standards in your field? Cryo-EM is emerging as an excellent tool for structural elucidation. The study discusses the application of this technique to define "one of the highest resolutions hitherto observed for a peptide fibril in cryo-EM". However, due to lack of proper definition of the peptide or its variant and the relevance of using the same at such high concentrations and the different pH values, makes it very confusing to gain a clear understanding of what the authors have to claim of such high-resolution structure.

Plectasin, like several other well-studied antimicrobial peptides have functions at the membrane interface of the microbes. Cryo-EM have been used to determine the structure of proteins in the membranous environment as well. Thus, with more and more growing interest on the structure-function correlation of biologically active peptides has shifted the focus to gaining structural insight into the functional interface of membranes for AMPs. Thus, a membrane mimicking environment or a representative system to define the functional interface would be much more relevant.

However, if the focus is to define a supramolecular structure that can release the monomers into the site of antimicrobial action upon pH reversal, it would require further experimentation to prove the same. Just a structural proof of the resultant peptides not being classical amyloid fibrils do not add much novelty in the ever growing field of structural biology.

- Is there enough detail provided in the methods for the work to be reproduced? Yes. The experimental details included are sufficient- although raise questions about the buffer conditions and peptide concentrations as already mentioned. Why do the authors choose to use the term in situ NMR is also unclear.

Reviewer #3 (Remarks to the Author):

This paper describes two cryo-EM structures of non-amyloid fibrils formed by a mutant form of an anti-microbial peptide (AMP) PPI42. The structure determination in this paper is decent and the models are reliable based on the resolution. However I am not sure about the scientific significance of this work because these fibrils were formed by an artificially designed mutant so that may not represent the general mechanism of the nature AMPs (please also see my major comment 1). So I will leave it up to the editor to decide whether this paper is a good fit to this journal. Here I provide some suggestions that may improve the manuscript:

Major comments:

1) The authors claim that these fibril structures may "serve as a basis for development of novel types of anti-microbial drugs" and "making a slow release of bioactive monomer possible", but in their results I did not see the benefit of forming fibrils instead of remaining as monomer, e.g. in Fig. 5B, the authors mentioned that PPI42 showed smaller clearing zones than wildtype. If the benefit of forming fibrils is increase stability and/or slow release, then is it possible to support this by some experiments? Such as whether the fibril form can increase the effective time against the bacterial, or the fibril form can resistant better to environment changes compared to wildtype?

2) In Introduction the authors mentioned that plectasin perform its activity by binding to lipid II, is the binding site known for PPI42? It should explain whether PPI42 is still active

in fibril form by check whether the binding site is blocked by fibril assembly.

3) In the first paragraph of Results, I did not find experiments to support the gel formation statements. Maybe the gel formation assays similar to that shown in Fig. 2 of Hughes et al., Science, 2018 should be added. Also in the third sentence starting with "When dialyzed against acidic pH...", please specify what is acidic pH (below 7? Or 5?)

4) Sometimes reversibility of gel formation does not necessarily correspond to reversibility of fibril formation. In the other words, the gel may dissolved in "acidic pH", but not necessarily the fibrils. AFM or EM showing fibrils are also disappeared after gel dissolved should be added.

5) The author can do a better job in describing the handedness of their structures. First of all, I believe the handedness in this study is correct, as there are alpha-helix in the structures, so maybe just a little clarification is needed. In the third paragraph of Results, the third sentence that start with "EM measurements revealed...", I believe negative stain EM cannot identify the handedness of the fibrils, as EM displays the projection of the object, unless extra experiment like platinum shadowing was performed. In the first paragraph under CryoEM structure of the supramolecular assembly subsection, the author claimed the helix within each fibril is left-handed, and I assume the handedness of the helix in the single fibril is also left-handed because the authors said the assembly of single fibril is almost identical with each fibril in superstructure. However the twist angles listed in the manuscript and Table S3 are all positive (e.g. 156.5 degrees in Table S3). In relion, positive angles means right-handed helix, so should the number actually be -156.5 degrees?

6) I am sorry but I find it difficult to follow the idea of double-strand helix within each fibrils of the superstructure (by the way, I think maybe protofilament is a better term for a single filament within the 2-fold fibrils). Where is the strand coming from, is the strand here means the monomers in one strand can be generated from each other by helical symmetry operation? In this case how can a double strand helix form by an ASU of seven monomers? Also maybe I missed in the manuscript, in the structure of single fibril, is the ASU still seven monomers of PPI42 or just one monomer (or two monomers each from one strand)?

7) Still the question about the seven-monomer-ASU, since the twist angle is around 150 degrees, but in the top view of Fig. S5A&B, the angles between two adjacent monomers seems to be ~50 degrees, maybe the authors can better label or color the figure to show which monomer is next to each other, and where are the corresponding helical rise and twist.

Minor comments:

1) In the second paragraph of Introduction, the authors mentioned cryo-EM as a tool to determine protein fibrils, but right after this sentence they mentioned three works that using X-ray. So it is better to change the position/sequence of these sentences.

2) Third paragraph in Results, the sentence start with "We observed frayed fibrils..." should be easier for the readers to follow if the author can cite Fig. 1B as well as "(indicated by) blue arrows".

3) Three paragraphs under subtitle "CryoEM structure of the supramolecular assembly" in Results should all have their own subtitle, as the first one is cryo-EM structure of the superstructure, and the second one is cryo-EM structure of the single fibril, and the third one is the comparison of wildtype and mutant.

Thank you again for submitting your manuscript "Reversible supramolecular assembly of the anti-microbial peptide plectasin into helical non-amyloid fibrils" to Nature Communications. We have now received reports from 3 reviewers and, on the basis of their comments, we have decided to invite a revision of your work for further consideration in our journal. Your revision should address all the points raised by our reviewers (see their reports below). Please note, while the reviewers find the cryo-EM structure of considerable interest, they question its physiological relevance and this caveat must be clearly stated and over-reaching conclusions need to be removed.

Thank you for your kind evaluation. We have now addressed all comments and suggestions by the reviewers. More specifically, we have addressed their concerns regarding the physiological relevance and have removed over-reaching conclusions.

REVIEWER COMMENTS

Reviewer #1 (Remarks to the Author):

Pohl et al described the first (to the best of my knowledge) CryoEM structure of a fibril-forming AMP. The structure described is of PPI42- plectasin variant D9S Q14K V36L, which forms fibrils more readily than the wt. Fibrillation is reversible in acidic conditions.

The structure is an assembly of the folded monomers, which is different from amyloid fibrils, and resembles actin fibrils in the aspect. DLS measurements indicated a pH and concentration-dependent equilibrium between monomers and fibrils. Direct monomer addition to the fibril is expected in a folded monomer assembly.

Antimicrobial activity showed a slight reduced activity of the mutant vs the wt, but it is unclear if the effect is related to fibrillation or to the mutations irrespectively. It is hard to control fibrillation conditions to answer this question. Histidine residues seems to serve as the control switch for pH-dependent fibrillation.

We have performed an array of further experiments to elaborate more detailed on some aspects. Specifically, the anti-microbial activity assay has been performed against a batch of different Gram-positive bacteria in solution. We could show that PPI42 is indeed more active against *Staphylococci* than the plectasin wildtype. We have also performed a titration study following light scattering, which captures the fibril formation to prove reversibility of the fibril formation.

Conceptual comments:

1. Since only the variant but not the wt plectasin form fibrils, it is hard to project the relevancy of this fibril formation in nature. Nevertheless, it can still promote the design of AMP supramolecular structures that provide stability with improved clinical and technological usage.

We thank this reviewer for their suggestion. We have changed the introduction of the paper to clarify the relevancy of this study, which is indeed connected with promoting new AMP variants that have beneficial properties due to their supramolecular assembly, as suggested.

2. Another aspect is lacking to indicate the functional relevancy of the self-assembly. Is the fibril only acting as a reservoir or the fibrils themselves have direct activity? This would not be a trivial assessment, and I am aware of the broad scope of experiments needed to answer this question.

We have analysed the binding site of the cell membrane and lipid 2, which is necessary for plectasin activity and found that it is not solvent accessible in the fibril form. We believe therefore, that the fibril can only serve as a reservoir but slow release upon dilution or lowering of pH is possible. We have addressed this in the results and discussion section of the paper.

Technical comment:

1. Intro: "Protein fibrils have been found to serve as structural elements in long-term memory in microorganisms^{1,2}." – I don't think that the memory is in the microbes (ref1 is SAA – a human protein ;ref 2 is *Aplysia* CPEB).

We thank this reviewer for their comment. These references and the statement have been removed from the Introduction.

2. I am confused about the variant: in the introduction it was mentioned: "Previously, we could show that the introduced mutations, which were originally implemented to increase anti-microbial activity, were predicted by MD simulations to only have a minor effect on the overall structure of plectasin₄₂ (Figure S1)". But the antibacterial activity shown here suggest reduced activity. This should probably be better addressed in the intro. Can it be related to the type of assay conducted? (see my comment #12 below).

The antimicrobial activity of plectasin PPI42 has been characterised and the new results including Minimal Inhibitory Concentration (MIC) against *Staphylococci* and *Streptococci* strains have been included in the manuscript. Results showed that PPI42 has improved potency against *Staphylococci* and retained activity against *Streptococci*. The results have been included in Figure 5.

3. Nomenclature is difficult- the two fibrils in the supramolecular structures are sometimes called fibrils and sometimes strands, which is especially confusing with β -strands. The convention, I think, is protofilament or protofibril, which assemble into "fibers" or "mature fibrils/fibers"

We agree with this reviewer, that our choice of nomenclature made the results difficult to follow. We have changed the nomenclature in the manuscript and named them protofilament and mature fibril.

4. Structure Table – I would add Molprobit and ClashScore percentiles.

Thank you for bringing this to our attention, we have added these numbers to the structure table.

5. Figure 2- explain the "N" and "I" in the figure legend. The figure is unclear regarding the arrangement of the individual monomers. Maybe color each monomer in a different color?

Figure 2 has been changed extensively to explain more clearly the arrangement of the monomers. The monomers of one asymmetric unit and each asymmetric have been colored individually, to give a better overview of the arrangement.

6. Figure S4 panel – it would be helpful to indicate on panel H the residues constituting the core (those indicated in panel F) for orientation.

Figure S4 has been changed and Panel H labels the residues in the hydrophobic core of the protofilament.

7. Figure 3 - The right and left panels show different clusters (and cluster 1 is not hydrophobic entirely, as it contains histidine) I suggest using two colouring schemes for the two clusters and trying to also show a joint view of the clusters in CPK. I am not sure, but maybe it will be easier if the structure would be shown as ribbons of secondary structure with the clusters in CPK. Also, the 90deg arrow is not indicative of the rotation. I am confused about panel E right: is panel B also showing more stabilizing interactions between the fibrils? Maybe a better coloring scheme will help with orientation here to distinguish between monomers and between protofibrils?

We thank this reviewer for their suggestion and changed Figure 3. We combined both clusters in a joined view now coloured in CPK. We found a representation as ribbons was difficult to distinguish between the clusters and the actual structure and decided to show the fibrils as cartoons and the clusters as spheres. The 90° arrow has been changed as well. Panel E has been changed and shows the interaction between the two protofibrils forming the mature fibril. We find that the current representation to be clearer.

8. The interface between the two protofibrils seems weak/small according to figure 3E. Was the solvent accessible surface area between monomers and between protofibrils calculated? It is often an indication of the compactness and relevancy of interfaces. It is directly correlated with free energy of binding.

We calculated the surface area, which indicates a tighter packing within one protofilament and weak interaction between the two protofilaments.

We have included this argument in the result section: 'Additionally, we calculated the buried area of PPI42 upon fibril formation. We found that upon formation of the protofilament the buried area increases around 28% from approx. 29% to approx. 57% with slight variation between the different monomers within the asymmetric unit (Table S1). The interfaces within the protofilament correspond to approx. 2500 Å² per monomer. The interface between the two protofilaments corresponds to 722 Å² per asymmetric unit, which corresponds to a small interface area for protein complexes. However, repetitive interfaces in a protein fibril naturally amplify the strength of a complex formation'

9. Figure 4- the structural differences can be related to the mutations but also the methods (crystal vs solution), as mentioned. Impossible to tell for sure. The two coordination of H18 in panel D should be mentioned in the legend. In the CryoEM, the single rotamer observed can be an issue of a lower resolution. The crystal structure of the mutant was not obtainable? If so, for what reason?

We agree with this reviewer, that using two different techniques for structure determination might lead to artifacts from the techniques. We were not able to crystallise PPI42. The reasons behind this are difficult to predict. The protein crystal of the plectasin wt were grown at pH 8.5. We suspect that under basic conditions fibril formation dominates over crystallization for PPI42, which might explain why we could not grow crystals for this variant. This hypothesis is further supported by our study following the fibril formation upon dilution into neutral pH using static light scattering. While PPI42 formed a hydrogel, confirming fibril formation, the plectasin wildtype showed protein crystal growth after 15 h (compare figure 5F and figure S8A).

We added a panel with the local resolution of the PPI42 monomer within the fibrils to Figure S4. Despite differences in local resolution, the amino acid stretch S9 to K14 still shows a resolution below 3 Å, which makes an unambiguous determination of the protein backbone possible. We believe the higher flexibility observed within this region originates from the lower coordination between the amino acids within the N-terminal loop. We checked the local resolution in both, the crystal structure and the cryo-EM structure again. In both structures the electron density is well defined and corresponds to a local resolution of 1.1 Å in the crystal structure and 2.5 Å in the cryo-EM structure. We believe therefore, that the two coordination of H18 observed for plectasin wt, but not for PPI42 is due to the Q14K mutations and not an issue of lower resolution. We have clarified that in the 'Result section'.

10. A view of the secondary structure is really missing for me to see that discussed interactions between protofibrils/monomers, and the overall packing. The one given in Figure S5 is not sufficient as it doesn't include the specific interactions discussed.

We agree with this reviewer's suggestion. Figure 3A has been changed and shows the fibrils in secondary structure now.

11. Figure S5 – I am lost in this figure. The two protofibrils in the superstructure do not look symmetric/equivalent. Is that possible in the asymmetric unit? what is the superimposition in panel B?

Figure S5 (now Figure S6) has been changed. Panels A to E show the 3D reconstruction and atomic model of the protofilament. In the protofilament structure, each plectasin monomer is related to the next by a strict helical symmetry. This relationship has been clarified in the figure now.

12. Antibacterial activity – it is plausible that in a case of a fibril-forming AMP, it is more indicative to conduct the assay in solution and not by diffusion.

The *in vitro* antimicrobial activity assays, MICs, were performed in solution and included in the manuscript. We believe the lower activity of PPI42 indicated in the diffusion assay is due to a general hinderance in the diffusion of PPI42. We have included this point in the results section.

13. Figure S6 –I don't understand the difference in the DLS results to the ones shown in figure 5.

Figure 5 shows the Scattering Intensity (%) which is related to the radius with the power of 6. Therefore, large particles such as aggregates or fibrils scatter proportionally more. Figure S6 (now S5) shows the %Mass of each fraction from the same measurement.

14. In my opinion, the indicated resolution is misleading, and in the case of CryoEM, it is important to indicate the range of resolution in the main text. Looking at the map fitting, it is clear to me that some areas are poorly fitted, which is fine, and acceptable, but the indication of salt bridges and other tight interactions, for example, should be mentioned with regard to the resolution of the local region.

We added the range of the local resolution for the reconstruction of the mature fibrils to the main text. We have also added an extra panel in figure S4E with a zoom on the local resolution of one PPI42 monomer from the fibril reconstruction to make it easier for the reader to follow.

Reviewer #2 (Remarks to the Author):

In their communication, "Reversible supramolecular assembly of the anti-microbial peptide (AMP) plectasin into helical non-amyloid fibrils" Christin Pohl et al. have studied the non-amyloid fibril formation by a mutant variant of the antimicrobial peptide plectasin-providing high-resolution structure using Cryo-EM.

- What are the noteworthy results?

The results are interesting and worth discussing in context of the structural insight obtained from Cryo-EM. However, to this reviewer it is unclear as to what is the background of the study. Lots of lacunae lies in terms of the functional implication of the entire structure elucidation. The introduction is elaborate, yet fails to lay the foundation for the study, the peptide or its variant. The major aim is not defined or established anywhere along the manuscript. Non-specific readers have to go back and forth with their previous manuscript in order to gain some understanding of the peptide or it's variant.

We thank this reviewer for their suggestion. We have added a paragraph to the introduction explaining the background and the aim of the study. We also introduced plectasin and the effort of creating new variants

- Will the work be of significance to the field and related fields? How does it compare to the established literature? If the work is not original, please provide relevant references.

Work is original and the detailed structural information as obtained from Cryo-EM is noteworthy, as it is a gradually expanding field in structure determination.

However, to this reviewer it seems important to correlate the structures to the peptide's functional attributes that, in turn, should enhance the readability and application of such structural studies. If not at the interface of peptide function, atleast an elaboration on the immediate application of such structural information is necessary, that appears to be largely missing from this study.

We have conducted the activity assay of PPI42 in solution to prove enhanced activity of PPI42 compared to the wildtype (especially against *Staphylococci*), which we find relevant in light of growing antibiotic resistance. The enhanced plectasin variant NZ2114 shows, similar to PPI42, enhanced activity against *Staphylococci*, which might be related to enhanced membrane binding through the replacement of D9 against a positively charged or non-charged amino acid. We believe that the hydrogel, containing fibrils may serve as a type of reservoir, that might enable slow release by dilution or change of pH. We have added these correlations of structure to function to the results and discussion section.

- Does the work support the conclusions and claims, or is additional evidence needed?

The work introduces AMPs and amyloids and goes on to define the structure of a mutant variant of plectasin

(without any direct introduction to this mutant, its function or plectasin as such) and using several biophysical methodologies proves that the pH-responsive self-assembly of PPI42 results in forming non-amyloid protein fibrils.

The authors suggest “supramolecular fibrils of plectasin may have the potential to provide a slow release of the monomer, which is known to be bioactive.” Given their study, this claim can only be a hypothesis and further experimentation would be necessary to claim this as an immediate application of their study (suggested in their discussions section). Moreover, the authors themselves have claimed that the fibrillar conformers have no effect on the functional potency of the peptide. Thus, it is difficult to gauge as to what is the claim that the authors want to put forward in their study.

We have added a paragraph to the introduction, introducing plectasin and the efforts made so far to develop new variants to tackle antibiotic resistance. We also added a section introducing pH responsive AMPs to support our hypothesis. We have clarified the advantages of reversible hydrogel formation might have for targeted therapy. The antimicrobial activity has been measured in solution now and we were able to show higher potency of PPI42 against *Staphylococci* compared to plectasin wildtype while maintaining similar potency against *Streptococci*. We believe therefore, this study might promote the design of new AMPs.

- Are there any flaws in the data analysis, interpretation and conclusions? - Do these prohibit publication or require revision?

While the structure determination in itself looks good enough, the study in whole seems to lack a defined implication altogether. Some of the major factors that are quite off-putting are as follows:

The study suggests that the peptide undergoes fibrillation in different pH conditions and peptide concentrations. This is already a known and established fact for several proteins and peptide systems. In fact high-resolution dynamic studies are also available to correlate the aggregation propensities of peptides. While several of these studies have indeed taken “amyloids” into consideration, the approach is relevant even for “non-amyloids.” Why is “non-amyloid” conformation of the peptide reinforced throughout the length of the manuscript, remains unclear, at least to this reviewer as to what does it signify?

We thank this reviewer for their suggestion and agree that reinforcing that the fibril formation is of non-amyloid nature is not necessary. We have changed the focus of the study towards the design of new AMPs and shortened the comparison between amyloid fibrils and PPI42 fibrils. We believe the main aim of this study and implication is defined in a better way now.

“Plectasin variant PPI42 formed a gel at peptide concentrations of approx. 5 mg/mL and higher, during dialysis at pH 5 and above” this too has been seen in several biologically active peptide systems. But the authors do not provide any introduction to this variant of plectasin and its antimicrobial attribute that might help in giving any understanding of the pH dependence of only the variant and not the wildtype.

We have now summarised the relevant findings of our previous article in the ‘introduction’ section to make it easier for the reader to follow our study. Additionally, we have added experimental data that show enhanced activity of the variant compared to the wildtype. We could show that fibrils also form at lower protein concentration. Based on our extensive biophysical characterisation we believe that PPI42 is present in an equilibrium between monomer and fibril which is shifted by protein concentration and pH. We additionally added an assay to evaluate the Gel elasticity, similar to the assay performed in Hughes et al., Science, 2018, which further supports this hypothesis.

The authors claim that the gel formation is irrespective of the chemical nature of the buffer system. However, it is very difficult to gain a clear picture as they have defined the use of different peptide concentrations and different buffer conditions for the different biophysical studies. It has been extensively shown in literature that a protein’s folding or fibrillation attribute is largely influenced by the pH and sample concentrations. Especially, with more and more literature being made available about protein’s liquid-liquid phase separation and gel formation that it is essential to define why a particular pH and peptide concentration has been used for a particular study and what the physiological implications are.

We added definitions throughout the manuscript, explaining the choice of buffer system and pH. Extensive testing of different buffer systems, which are described in the manuscript and SI (e.g. AFM measurements in different buffer systems), showed that the fibril formation and morphology was not influenced by different buffer systems but depended on pH and protein concentration. Some methods (e.g. CD) required low protein concentrations. Consequently, the pH-onset for fibril formation was higher meaning we had to change buffer systems to ensure good buffering capacity. We have added the exact conditions for all measurements to the figure captions and added explanations to the choice of buffer system for each measurement to the ‘Materials and Methods’ section.

For an antimicrobial peptide (AMP), a peptide concentration of 5mg/ml appears to be too high, thus lowering the implication of the structural insight into the fibrillar forms. Is this concentration even relevant to the peptides function? What is its effective microbial inhibitory value? Has the authors performed any experiment to determine the same?

It seems only logical that the authors provide atleast some data on the cytotoxicity at the concentrations used for the fibrillar structure determination. Although it is known from literature that protein fibrils are less cytotoxic as opposed to the intermediate oligomers; however, this is not always true, since fibrils often interfere with the

homeostatic conditions. Therefore, even the clinical implications suggested in the discussion has to be tested at the used concentrations against the host cellular systems, while providing antimicrobial data for the same.

To address this reviewer's comments, we have looked into data produced during the development stage of the peptide by Novozymes A/S. We found that stock solutions of PPI42 ranging from 2 mg/ml up to 5 mg/ml in 10 mM phosphate buffer pH 4 with 9 mg/mL NaCl were used in a series of *in vitro* cytotoxicity assays, *in vivo* toxicity studies in mice and rats and efficacy studies in mice using a peritonitis model against *Staphylococcus aureus*. Hence, a stock solution of 5 mg/mL has been shown to be relevant to the protein function. We agree with this reviewer that a protein concentration of 5 mg/mL is higher than orally given or injected. A possible application for this high protein concentrations is, however, a direct application on the skin. As this application is purely speculative and beyond the scope of this study, we have not included this into the manuscript.

Some of the results were presented at the ICAAC Interscience Conference on Antimicrobial Agents and Chemotherapy Sept 27-30, 2006 (poster F2-1166) (see Figure 3 below).

Previously performed *in vitro* cytotoxicity, *in vivo* toxicity and efficacy studies

Both Plectasin wt and PPI42 were tested for *in vitro* cytotoxicity in a Red Blood Cell Hemolysis assay and both compounds were found to be non-toxic up to a concentration of 1024 µg/mL eliciting a haemolytic effect 0.8 – 4.5%.

Additionally, the *in vivo* toxicity of PPI42 compound was evaluated in female NMRI mice. A single intravenous bolus injection of 50 mg/kg was administered. The mice were observed for clinical signs for a period of 4 days after dosing, after which they were euthanized. All mice were subjected to a macroscopic evaluation and relevant organs weights were recorded. It was concluded that the No Observed Adverse Effect Level (NOAEL) and The Maximum Tolerated Dose (MTD) for PPI42 molecule was > 50 mg/kg.

Furthermore, the *in vivo* efficacy of plectasin PPI42 variant and wild type molecule was determined in the murine peritonitis infection model. In brief, the dose-response curves were investigated following either iv or sc administration of a single dose of test item ranging from 0.08 – 20 mg/kg and the Effective Dose (ED50) was determined. The effect was tested against the methicillin sensitive *Staphylococcus aureus* (MSSA) E33235 strain. Female NMRI mice were infected at T= -1 hour with ~2.5E7 CFU of the *S. aureus* test strain and treated with a single dose of a given plectasin molecule at T = 0 hours. 4 hours later, the mice were sacrificed and the CFU determined in blood and peritoneal fluid.

The dose-response curves for plectasin PPI42 variant against *S. aureus* showed a reduction in colony counts compared to start of treatment in the dose range of 0.63 – 20 mg/kg, thus PPI42 plectasin variant was very effective against the tested strain of *Staphylococcus aureus* (figure 1 and 2). The ED50 of PPI42 was 0.5 mg/kg in peritoneum and 0.3 mg/kg in blood. The corresponding ED50 values for the efficacy of plectasin wildtype against the same test strain was found to be 3.5 mg/kg in peritoneal fluid and 5.8 mg/kg in blood (table 1). The MIC of PPI42 variant against the test *S. aureus* E33235 strain was 0.2 µg/ml and 4 µg/ml for plectasin wild type, thus correlates with the improved *in vivo* performance of PPI42 variant. The efficacy of the PPI42 variant was also tested against two methicillin resistant *S. aureus* strains (MRSA) using the same animal model, and it was concluded that treatment with PPI42 variant also resulted in a reduction in CFU's compared to start of treatment, thus, showing that PPI42 was effective against the two MRSA strains (data not shown).

Figure 1: Dose Response curves. Effect of single dose treatment with plectasin wild type and PPI42 variant against *S. aureus* E33235 at 4 hours after treatment in peritoneal fluid in a murine peritonitis model.

Figure 1: Dose Response curves. Effect of single dose treatment with plectasin wild type and PPI42 variant against *S. aureus* E33235 at 4 hours after treatment in peritoneal fluid in a murine peritonitis model.

Figure 2: Dose Response curve. Effect of single dose treatment with plectasin wildtype and PPI42 against *S. aureus* E33235 at 4 hours after treatment in blood in a murine peritonitis model.

Table 1: ED50 in mg/kg tested against *S. aureus* E33235 of plectasin wildtype and PPI42.

	Plectasin WT	Plectasin PPI42
Peritonitis / ED50	3.5 (peritoneal fluid)	0.5 (peritoneal fluid)
	5.8 (blood)	0.3 (blood)

High Throughput Screening Of Antimicrobial Peptides: In Vitro and In Vivo Optimization of Plectasin

D. Raventos, PH. Mygind, D. Sandvang, L. Nielsen, BT. Ravn, NK. Soni, S. Buskov, J. Lichtenberg, P. Oestergaard, DA Skovlund, MV Sørensen, MT Hansen, B. Christensen, BE. Christensen & H-H. Kristensen
Novozymes A/S, Bagsvaerd, Denmark

Contact information:
HaHK@novozymes.com
Phone: +45 44421823
Mobile: + 45 30791823

Introduction

Antimicrobial peptides are a recently discovered group of antimicrobial agents. They are simple peptides that are widely distributed in animals and plants and collectively show activity against a broad range of microbial pathogens. They have a number of characteristics that make them interesting candidates for pharmaceutical development. Notably they are fast-acting, killing the microorganisms rather than inhibiting them, and with little observed resistance development. Plectasin is a newly discovered defensin-type antimicrobial peptide isolated from the saprophytic ascomycete fungus *Pseudoplectania nigrella*. Plectasin is potent active against several Gram-positive bacteria including drug-resistant strains such as MRSA and VRSA. Here, a mutational campaign and high throughput screening (HTS) setup aiming at increasing the *in vitro* and *in vivo* potency of wt plectasin (NZZ2000) against *S. aureus* will be presented.

Screening setup

The screening setup was divided in a primary screen, a secondary screen and a tertiary screen. The primary screen, a simple plate screen against *S. aureus*, included a total of 660 000 plectasin variants. The increased activity of the selected variants was confirmed in a plate assay against both *S. aureus* and *S. pneumoniae*, where the activity was quantified based on clearing zone size and the most potent variants progressed to the secondary screen.

The secondary screen triaged purified and quantified material of 120 variants against 14 *Staphylococci* and *Streptococci*.

The tertiary screen qualified the 17 most potent candidates in more than 15 different *in vitro* and *in vivo* assays.

Figure 1. Schematic setup of HTS

Primary Screen - 660 000 variants

- plate screen vs. *S. aureus*
- retest vs. 2 x *staph* & 2 x *strep*

Secondary Screen - 120 variants

- MICs vs. 14 *staph* & *strep*

Tertiary Screen - 17 variants

- a range of *in vitro* & *in vivo* parameters

In vivo profiling - 1 lead + 2 backups

Minimal inhibitory concentration (MIC)

The MIC of selected variants was determined by the microbroth dilution method using cationic-adjusted MHB according to the protocol of NCCLS / CLSI.

A total of 60 *S. aureus* and 50 *Streptococci* were included.

Figure 2. MICs of selected candidates from the 3 screen vs. *Staphylococci* and *Streptococci*.

In vitro cytotoxicity

The *in vitro* cytotoxicity was assayed using a human red blood cell test and a cytotoxicity test on a murine cell line.

Figure 3. In vitro cytotoxicity of selected variants

In vivo toxicity

The *in vivo* toxicity of each of the 3 screen compounds were evaluated in 4 NMR1 female mice. A single IV dose of 50 mg/kg was administered. Clinical signs, body and organ weights were followed for 4 days while macroscopic pathology was investigated at necropsy.

Compound	NOAEL Mg/kg	MTD mg/kg
NZZ2000	>50	>>> 50
NZZ2097	>50	>>> 50
NZZ2098	>50	>>> 50
NZZ2076	< 50	>> 50
NZZ114	< 50	> 50
NZZ116	< 50	>> 50

In vivo efficacy

The *in vivo* efficacy of selected variants was determined in a murine peritonitis infection model. Mice were infected at T = -1 hour with $\sim 2.5 \times 10^7$ CFU of *S. aureus* and treated with a single dose of a given variant at T = 0 hours. 4 hours later, the mouse was sacrificed and the CFU determined in blood and peritoneal fluid. For each variant, the following doses were administered to three mice: 20 mg/kg, 10 mg/kg, 5 mg/kg, 2.5 mg/kg, 1.25 mg/kg, 0.63 mg/kg, 0.31 mg/kg, 0.16 mg/kg and 0.08 mg/kg and the ED50 was determined.

Figure 4. ED50 in mg/kg and corresponding 95% confidence interval tested against *S. aureus* E33235

Conclusion

A total of 660 000 plectasin variants have been screened in the presented campaign. The setup consisted of a primary, secondary and tertiary screen each progressively increasing the level of characterization while decreasing the number of tested variants.

The most potent variants exhibited an 8-fold increase in the MIC against *Staphylococci* while maintaining the original potency against *Streptococci*. Other *in vitro* parameters such as MIC/MBC, MPC and frequency of spontaneous resistance (<2 E-1) remained unaltered.

In vivo, the described mutants all exhibited an increased potency and showed up to a 6-8 fold increase in potency in the murine peritonitis model.

The data show that plectasin can be optimized to exhibit potent activity *in vitro* and *in vivo* against *Staphylococci* while retaining properties that make it an attractive candidate for development into a human therapeutic. Selected lead candidates - equally potent *in vitro* and *in vivo* in selected animal models to antibiotics such as Vancomycin, Cubicin and Linezolid - have entered further preclinical characterization.

Figure 3: Poster presented at the ICAAC Interscience Conference on Antimicrobial Agents and Chemotherapy Sept 27-30, 2006 (poster F2-1166). Plectasin wt is referred to as NZZ2000; NZZ2098 corresponds to the PPI42 variant.

The study suggests "significant structural difference in the N-terminal loop region between amino acids 9 and 14". It is possible to find a direct correlation between this flexible loop regions to the pH sensitivity of the distinct fibrillar pathway for the variant using heteromolecular 2D NMR. More so, applying experiments to gain atomic resolution into the dynamics of the early folding events might be helpful in gaining the differential aggregation pathway, particularly, in the different pH systems.

Regarding this point, we are not entirely sure what the reviewer refers to by 'heteromolecular 2D NMR'. Despite the lack of isotope-enriched peptide, we were able to conduct a full assignment of backbone amide groups (^1H - ^{15}N groups) at natural isotopic enrichment using cryogenically cooled detection electronics and high-field NMR instrumentation for both wildtype and PPI42. The chemical shift changes between wildtype and PPI42 at 298 K are summarized in the chemical shift mapping shown below and added to the Supporting information as figure S10. We concur that the observed changes shed additional light onto fibrillation events, and thus support the primary focus of the current study.

The three mutated residues are marked by asterisks in Figure S10 and unsurprisingly these residues show some of the largest changes in chemical shifts between wildtype and PPI42. In addition, conformational changes between the mutated residues 9 and 14 are evident especially for E10 and D11. These residues will be positioned at the monomer interface between the hydrophobic patches formed by M13/H16 and F2/H18 on neighbouring monomers upon fibrillation. Adjacent to M13/H16 are further K26 and G28, which accordingly also show significant changes in local structure and/or dynamics between PPI42 and wildtype. We concur that the mechanistic role of these residues in nucleation could be further scrutinized through heteronuclear relaxation and relaxation dispersion experiments in NMR studies dedicated to the dynamic changes to peptide chain upon nucleation and fibrillation. The last residues with strong changes in backbone chemical shifts are C4 and C30, indicating that disulfide bond conformations may differ between the two constructs.

Hence, heteronuclear NMR indeed identifies differences in local structures between fibril-forming and non-fibril-forming construct for the N-terminal loop region, residues adjacent to the histidine-containing hydrophobic

patches, and some core cysteine residues. A paragraph, discussing these results has been added to the 'Supplementary Discussion'.

- Is the methodology sound? Does the work meet the expected standards in your field?

Cryo-EM is emerging as an excellent tool for structural elucidation. The study discusses the application of this technique to define "one of the highest resolutions hitherto observed for a peptide fibril in cryo-EM". However, due to lack of proper definition of the peptide or its variant and the relevance of using the same at such high concentrations and the different pH values, makes it very confusing to gain a clear understanding of what the authors have to claim of such high-resolution structure.

We have redefined the overall aim of the study (see previous comments) in the 'Introduction' section and discussed possible application in the "Discussion" section.

Plectasin, like several other well-studied antimicrobial peptides have functions at the membrane interface of the microbes. Cryo-EM have been used to determine the structure of proteins in the membranous environment as well. Thus, with more and more growing interest on the structure-function correlation of biologically active peptides has shifted the focus to gaining structural insight into the functional interface of membranes for AMPs. Thus, a membrane mimicking environment or a representative system to define the functional interface would be much more relevant.

We agree with this reviewer that discussion of the structure-function relationship is very relevant for this study. We have added a paragraph to the 'Results' sections, where we discuss the structure-function relationship in relation to the activity in fibril form. We believe PPI42 is not active in fibrillar form (figure S7), as the membrane binding residues, as well as the lipid 2 binding residues are buried in the core of the protofilaments. However, the fibril as a reservoir and slow-releasing AMP is a possible application. We have discussed this point in the 'Discussion' section. We agree that a membrane mimicking environment would shine light on future development of PPI42 as a protein drug. However, this is beyond the scope of this study.

However, if the focus is to define a supramolecular structure that can release the monomers into the site of antimicrobial action upon pH reversal, it would require further experimentation to prove the same. Just a structural proof of the resultant peptides not being classical amyloid fibrils do not add much novelty in the ever growing field of structural biology.

We have added a titration study where we followed the fibril formation upon dilution into neutral pH using static light scattering and subsequently titrated with HCl. The recovery of the monomer was confirmed by NMR (figure 5F and figure S8B). Furthermore, we could show, using NMR spectroscopy that the equilibrium between fibril and monomeric state is concentration dependent. We believe therefore, that PPI42 has the potential for slow release of the active monomer used in targeted anti-microbial treatment.

- Is there enough detail provided in the methods for the work to be reproduced?

Yes. The experimental details included are sufficient- although raise questions about the buffer conditions and peptide concentrations as already mentioned. Why do the authors choose to use the term in situ NMR is also unclear.

We have added details and reasoning for the choice of buffer systems, pH and protein concentration to each technique used in this study in the 'Material and Methods' section.

The term "in situ NMR" had originally been used to indicate that dynamic changes in a reaction system were monitored; we concur that the observation of fibrillation may equally well omit the "in situ" phrase and the manuscript has been amended accordingly. Thank you for bringing this point to our attention.

Reviewer #3 (Remarks to the Author):

This paper describes two cryo-EM structures of non-amyloid fibrils formed by a mutant form of an anti-microbial peptide (AMP) PPI42. The structure determination in this paper is decent and the models are reliable based on the resolution. However I am not sure about the scientific significance of this work because these fibrils were formed by an artificially designed mutant so that may not represent the general mechanism of the nature AMPs (please also see my major comment 1). So I will leave it up to the editor to decide whether this paper is a good fit to this journal. Here I provide some suggestions that may improve the manuscript:

Major comments:

1) The authors claim that these fibril structures may "serve as a basis for development of novel types of anti-microbial drugs" and "making a slow release of bioactive monomer possible", but in their results I did not see the benefit of forming fibrils instead of remaining as monomer, e.g. in Fig. 5B, the authors mentioned that PPI42 showed smaller clearing zones than wildtype. If the benefit of forming fibrils is increase stability and/or slow release, then is it possible to support this by some experiments? Such as whether the fibril form can increase the effective time against the bacterial, or the fibril form can resistant better to environment changes compared to wildtype?

We have added a paragraph to the 'Introduction' section of the manuscript, where we introduce the field of tunable AMP self-assembly and the connection to targeted anti-microbial therapy. We have also focused on defining the aim of our study, which we believe is easier to follow for the reader. Additionally, we have performed

further experiments, which support the benefit of the fibril formation in the context of slow release. By analysing the position of the active site within the fibril structure, we believe that PPI42 is not active in its fibrillar form (figure S7). Following the fibril formation upon dilution into neutral pH and subsequent acid titrations, we could show that the fibril formation is completely reversible (figure 5F and S8B). Our NMR measurements showed that the equilibrium is pH and protein concentration dependent. Therefore, we believe that slow-release of the active monomer either by change of pH or dilution is possible for PPI42.

2) In Introduction the authors mentioned that plectasin perform its activity by binding to lipid II, is the binding site known for PPI42? It should explain whether PPI42 is still active in fibril form by check whether the binding site is blocked by fibril assembly.

We thank this reviewer for their suggestion. We have analysed the binding site for the bacterial membrane and lipid 2, which has been determined for the plectasin wildtype by Schneider et al. (2010) *Science*. Due to the high structural similarity between PPI42 and wildtype, we believe that similar residues are involved in the membrane and lipid 2 binding for PPI42. Our results indicate that PPI42 is most likely not active in its fibrillar form because both binding sites are located in the core of the protofilament (figure S7). We have added this result to the 'Results' section of the manuscript and discussed it further in the 'Discussion' section.

3) In the first paragraph of Results, I did not find experiments to support the gel formation statements. Maybe the gel formation assays similar to that shown in Fig. 2 of Hughes et al., *Science*, 2018 should be added. Also in the third sentence starting with "When dialyzed against acidic pH...", please specify what is acidic pH (below 7? Or 5?)

Again, we thank the reviewer for their suggestion, which we found has enhanced the manuscript significantly. We have performed a gel formation assay similar to the one shown in Hughes et al. (2018) *Science* to make it easier to visualise the formation of the hydrogel. Additionally, we specified the exact pH, we referred to for each measurement.

4) Sometimes reversibility of gel formation does not necessarily correspond to reversibility of fibril formation. In the other words, the gel may dissolved in "acidic pH", but not necessarily the fibrils. AFM or EM showing fibrils are also disappeared after gel dissolved should be added.

We appreciate this reviewers concern regarding the reversibility of the fibril formation, which can indeed be due to dilution but not necessary reversibility. We have followed the fibril formation upon dilution into neutral pH (pH 7) by static light scattering and subsequently titrated the sample with HCl until the scattering intensity was equivalent to the scattering intensity at the start. We measured the NMR signal of the protein, which only captures the monomeric protein at the start, at the equilibrium after dilution and after the acid titration.

Light scattering assay was performed. These measurements confirmed that the fibril formation is indeed reversible upon change of pH. We found that these 'batch' measurements followed the reversibility in a better way than confirming reversibility based on non-visible monomers in AFM or EM.

5) The author can do a better job in describing the handedness of their structures. First of all, I believe the handedness in this study is correct, as there are alpha-helix in the structures, so maybe just a little clarification is needed. In the third paragraph of Results, the third sentence that start with "EM measurements revealed...", I believe negative stain EM cannot identify the handedness of the fibrils, as EM displays the projection of the object, unless extra experiment like platinum shadowing was performed. In the first paragraph under CryoEM structure of the supramolecular assembly subsection, the author claimed the helix within each fibril is left-handed, and I assume the handedness of the helix in the single fibril is also left-handed because the authors said the assembly of single fibril is almost identical with each fibril in superstructure. However the twist angles listed in the manuscript and Table S3 are all positive (e.g. 156.5 degrees in Table S3). In relation, positive angles means right-handed helix, so should the number actually be -156.5 degrees?

To the negative stain comment:

We agree with the reviewer that the handedness cannot be determined by negative stain and therefore, we removed any reference to handedness in the negative stain paragraph of the main text.

About the cryo-EM handedness:

We agree with the reviewer that the handedness of both cryo-EM structures (the mature fibrils composed of two protofilaments and the isolated protofilament) was poorly described in the original manuscript. We have now extensively revised both the main text and the figures 2 and S5, which describe the cryo-EM 3D reconstructions of the mature fibril and the protofilament of plectasin respectively.

The protofilaments in isolated form and in the mature fibril form right-handed helical assembly but the structures differ in their helical parameters due to a curvature in the protofilament upon forming the mature fibril. For the mature fibril, the axial rise is 25.2 Å and the twist is 15.8°. The monomers in the asymmetric unit of the mature fibrils arrange in a near-helical way with an axial rise of $3.75 \pm 0.15^\circ$ and an average twist of $156.48 \pm 2.25^\circ$. For the isolated protofilament the axial rise is 3.76 Å and the twist is 156.5° and the monomer arrangement is strictly helical.

6) I am sorry but I find it difficult to follow the idea of double-strand helix within each fibrils of the superstructure (by the way, I think maybe protofilament is a better term for a single filament within the 2-fold fibrils). Where is the strand coming from, is the strand here means the monomers in one strand can be generated from each other by helical symmetry operation? In this case how can a double strand helix form by an ASU of seven monomers?

Also maybe I missed in the manuscript, in the structure of single fibril, is the ASU still seven monomers of PPI42 or just one monomer (or two monomers each from one strand)?

7) Still the question about the seven-monomer-ASU, since the twist angle is around 150 degrees, but in the top view of Fig. S5A&B, the angles between two adjacent monomers seems to be ~50 degrees, maybe the authors can better label or color the figure to show which monomer is next to each other, and where are the corresponding helical rise and twist.

We agree with this reviewer that our description of the cryo-EM structures was not consistent and we have changed it. We use the terms protofilament and mature fibril to describe the single filament and the superstructure consistently throughout the manuscript. We agree that the term strands was confusing in the context of the overall structure, especially because the monomers within are not generated within a helical symmetry operation. The interaction of the two protofilaments forming the mature fibril (previously described as superstructure), results in a twist within the protofilament. Therefore, the ASU is composed of seven plectasin monomers which are different to each other. The monomers in the asymmetric unit of the mature fibrils arrange in a near-helical way with an axial rise of 3.75 ± 0.15 Å and an average twist of $156.48 \pm 2.25^\circ$. Each protofilament is a single right-handed helix, which we previously described as strands. In the mature fibril, the axial rise is 25.2 Å and the twist is 15.8° between the different ASUs. In the isolated protofilament structure, the ASU is a single plectasin monomer. The structure of the isolated protofilament shows a right-handed helix with an axial rise of 3.76 Å and a twist of 156.5° . The alignment of the protofilament from the mature fibril and with the isolated protofilament, showed that the seven monomers of plectasin which are composing the ASU in the mature fibril are arranged in a similar way as the strict helical arrangement of monomers in the structure of the isolated protofilament (all-atom RMSD 0.727 Å).

We have changed the main text ('results' section) and figure 2 and S5 to make this arrangement clearer to the reader.

Minor comments:

1) In the second paragraph of Introduction, the authors mentioned cryo-EM as a tool to determine protein fibrils, but right after this sentence they mentioned three works that using X-ray. So it is better to change the position/sequence of these sentences.

Thank you for bringing this to our attention, we have rewritten this part of the introduction.

2) Third paragraph in Results, the sentence start with "We observed frayed fibrils..." should be easier for the readers to follow if the author can cite Fig. 1B as well as "(indicated by) blue arrows".

We agree with this reviewer's suggestion and have referred to the arrows shown in the figure also in the main text.

3) Three paragraphs under subtitle "CryoEM structure of the supramolecular assembly" in Results should all have their own subtitle, as the first one is cryo-EM structure of the superstructure, and the second one is cryo-EM structure of the single fibril, and the third one is the comparison of wildtype and mutant.

We have added additional headings to the results part to separate the paragraphs in a better way, as this reviewer rightfully pointed out.

REVIEWERS' COMMENTS

Reviewer #1 (Remarks to the Author):

In their revisions, Pohl et al comprehensively addressed my comments. The addition of analyses to show functional relevance is contributing to significance, and the modified figures are much better and provide great service to appreciating the beauty of the structure.

Reviewer #2 (Remarks to the Author):

I am satisfied with the revised version.

Reviewer #3 (Remarks to the Author):

The revised manuscript addressed most of my previous concerns and better explained the physiological relevance of this study. So, I think I am favorable for this study to publish on Nature Communication. Meanwhile, some editions/clarifications can be made to further improve the manuscript.

Major comments:

- 1) The authors did a very good job in revising Figure 2, and the new version really helps the audiences to understand the structure of PPI42 fibrils. However, the quality of Figure 3 can be further improved, as well as the description between line 163-179 of the revised manuscript. More specifically, I think the authors should better explain the relative position of monomer i , $i+1$, $i-1$, n , m , etc. On amyloid fibrils $i/i-1/i+1$ usually refer to three consecutive layers that stack on top of each other to form fibrils, and since PPI42 fibril structure is more complicated than regular amyloid fibrils, I am wondering whether the authors can label relative positions of $i/i-1/i+1/n/n+1/m$ on a figure that similar to Figure 2A and C. For example, if we define $au3_3$ as i , then is $i+1$ $au3_4$, $au3_5$, or $au4_3$? And in this case which monomer is n and m ?
- 2) For the C3 symmetry of isolated protofilament, the authors can add more details in the Methods and/or main text to describe how they found the isolated protofilament has a 3-fold symmetry before they applied this symmetry into data processing. And if I understand it correctly, a C3 symmetry means the fibrils can be divided into three "strands" that symmetrically related to each other with a 120° rotation. In this case can the authors add a panel in Figure S6 with the same view of panel A but paint these three strands in three different colors? It will help the audiences to better understand the C3 symmetry of the isolated protofilament.
- 3) In this study, the authors found that the fibril formation of PPI42 is pH and protein concentration dependent, and to me the concentration dependence is even more important than pH dependence since according to the authors the concentration dependence is linked to slow release of functional PPI42. In this case I suggest adding concentration dependence in the title, abstract, and first sentence of Discussion of this manuscript.

Minor comments:

- 1) Line 101, better to reference Figure 1A at the end of this sentence.
- 2) Line 145 and 154, the twist angle here is 156.48° but the label in Figure 2 is 156.5°, if they are referring the same angle, better to make them consistent. Similarly on line 151 the angle is 15.75° whereas on Figure 2 is 15.8°.
- 3) Line 160, does the "fibril center" means "protofilament center" here?
- 4) Line 166-175, the authors were citing Figure 3D first, and then Figure 3A-3B-3C, it is better to label the figure panel as the same sequence they appeared in the main text. Similarly on line 233, I think

authors cite Figure S10 before S7,8,9. Also in line 317, Figure S5D is cited after Figure S5F.
5) Line 311, the authors cited Figure S7 here, but I think they meant Figure S8B.

Reviewer #1 (Remarks to the Author):

In their revisions, Pohl et al comprehensively addressed my comments. The addition of analyses to show functional relevance is contributing to significance, and the modified figures are much better and provide great service to appreciating the beauty of the structure.

We thank this reviewer for their evaluation and kind words.

Reviewer #2 (Remarks to the Author):

I am satisfied with the revised version.

We thank this reviewer.

Reviewer #3 (Remarks to the Author):

The revised manuscript addressed most of my previous concerns and better explained the physiological relevance of this study. So, I think I am favorable for this study to publish on Nature Communication. Meanwhile, some editions/clarifications can be made to further improve the manuscript.

Major comments:

1) The authors did a very good job in revising Figure 2, and the new version really helps the audiences to understand the structure of PPI42 fibrils. However, the quality of Figure 3 can be further improved, as well as the description between line 163-179 of the revised manuscript. More specifically, I think the authors should better explain the relative position of monomer i , $i+1$, $i-1$, n , m , etc. On amyloid fibrils $i/i-1/i+1$ usually refer to three consecutive layers that stack on top of each other to form fibrils, and since PPI42 fibril structure is more complicated than regular amyloid fibrils, I am wondering whether the authors can label relative positions of $i/i-1/i+1/n/n+1/m$ on a figure that similar to Figure 2A and C. For example, if we define $au3_3$ as i , then is $i+1$ $au3_4$, $au3_5$, or $au4_3$? And in this case which monomer is n and m ?

We agree with this reviewer that the quality of Figure 3 could be further improved. We have revised this figure extensively and adjusted the numbering system to the relative positions in the asymmetric unit, established in Figure 2 and eliminated the numbering with letters i , n and m . We think this improves the understanding of the reader significantly.

2) For the C3 symmetry of isolated protofilament, the authors can add more details in the Methods and/or main text to describe how they found the isolated protofilament has a 3-fold symmetry before they applied this symmetry into data processing. And if I understand it correctly, a C3 symmetry means the fibrils can be divided into three "strands" that symmetrically related to each other with a 120° rotation. In this case can the authors add a panel in Figure S6 with the same view of panel A but paint these three strands in three different colors? It will help the audiences to better understand the C3 symmetry of the isolated protofilament.

We are not entirely sure what this reviewer means with C3 symmetry of the isolated protofilament. The isolated protofilament has a helical symmetry and but no axial symmetry (C1). Figure S6 shows three consecutive monomers related by an azimuthal angle of 156.5° and an axial rise of 3.76 Å. However, the isolated protofilament does not, consists of three strands related to each other by a 120° rotation.

3) In this study, the authors found that the fibril formation of PPI42 is pH and protein concentration dependent, and to me the concentration dependence is even more important than pH dependence since according to the authors the concentration dependence is linked to slow release of functional PPI42. In this case I suggest adding concentration dependence in the title, abstract, and first sentence of Discussion of this manuscript.

We have added the concentration dependency of the fibril formation to the title and abstract to enhance the importance of it for the physiological relevance of PPI42.

Minor comments:

1) Line 101, better to reference Figure 1A at the end of this sentence. We thank this reviewer for pointing out, that a figure reference would be beneficial here. We have added a Figure panel (Figure SX) to the supplementary showing the reversibility after dialysis. Figure 1A shows a dilution of the PPI42 stock solution into neutral pH, but not the reversibility of the gel formation.

2) Line 145 and 154, the twist angle here is 156.48° but the label in Figure 2 is 156.5°, if they are referring the same angle, better to make them consistent. Similarly on line 151 the angle is 15.75° whereas on Figure 2 is 15.8°. This has been changed in the manuscript and we consistently refer to 2 digits now.

3) Line 160, does the "fibril center" means "protofilament center" here? This reviewer is correct in pointing out that we refer to the center of the protofilament in this case. This has been changed in the manuscript.

4) Line 166-175, the authors were citing Figure 3D first, and then Figure 3A-3B-3C, it is better to label the figure panel as the same sequence they appeared in the main text. Similarly on line 233, I think authors cite Figure S10 before S7,8,9. Also in line 317, Figure S5D is cited after Figure S5F.

5) Line 311, the authors cited Figure S7 here, but I think they meant Figure S8B. Thank you for bringing this to our attention. This has been changed in the manuscript.